JCB Journal of Cell Biology

# Polydom/SVEP1 binds to Tie1 and promotes migration of lymphatic endothelial cells

Ryoko Sato-Nishiuchi[1], Masamichi Doiguchi[1], Nanami Morooka[1,2], and Kiyotoshi Sekiguchi[1]

Polydom is an extracellular matrix protein involved in lymphatic vessel development. Polydom-deficient mice die immediately after birth due to defects in lymphatic vessel remodeling, but the mechanism involved is poorly understood. Here, we report that Polydom directly binds to Tie1, an orphan receptor in the Angiopoietin–Tie axis, and facilitates migration of lymphatic endothelial cells (LECs) in a Tie1-dependent manner. Polydom-induced LEC migration is diminished by PI3K inhibitors but not by an ERK inhibitor, suggesting that the PI3K/Akt signaling pathway is involved in Polydom-induced LEC migration. In line with this possibility, Akt phosphorylation in LECs is enhanced by Polydom although no significant Tie1 phosphorylation is induced by Polydom. LECs also exhibited nuclear exclusion of Foxo1, a signaling event downstream of Akt activation, which was impaired in Polydom-deficient mice. These findings indicate that Polydom is a physiological ligand for Tie1 and participates in lymphatic vessel development through activation of the PI3K/Akt pathway.

## Introduction

Polydom, also known as SVEP1, is an ECM protein >300 kD that consists of multiple domains, including a von Willebrand factor type A domain, a pentraxin domain, 10 EGF-like domains, and an array of complement control protein (CCP) domains. Polydom is predominantly expressed in embryonic mesenchymal tissues such as those in the lung, stomach, and intestine, and has the ability to bind to integrin α9β1 (Sato-Nishiuchi et al., 2012). Polydom-deficient mice showed severe edema and died immediately after birth due to respiratory failure (Karpanen et al., 2017; Morooka et al., 2017), a phenotype indicative of defects in lymphatic development. In Polydom-deficient mice, the primitive lymphatic plexus was formed but failed to undergo remodeling, such as sprouting of new capillaries and formation of collecting lymphatic vessels (Morooka et al., 2017). Consistent with a role in lymphatic vessel remodeling, Polydom is deposited around lymphatic endothelial cells (LECs), although it is predominantly expressed in mesenchymal cells that express platelet-derived growth factor receptor-α (Morooka et al., 2017).

Similar defects in lymphatic development have been reported in knockout mice for other proteins, including Tie1 (D'Amico et al., 2010; Puri et al., 1995; Qu et al., 2010; Sato et al., 1995; Shen et al., 2014), EphrinB2 (Mäkinen et al., 2005; Wang et al., 2010), Angiopoietin-2 (Ang2; Dellinger et al., 2008; Gale et al., 2002), Podoplanin (Schacht et al., 2003), and Foxc2 (Norrmén et al., 2009; Petrova et al., 2004), with all of these mice exhibiting failure of lymphatic vessel remodeling. Null mice for integrin α9, a known receptor for Polydom, displayed chylothorax and defects in lymphatic valve formation but did not phenocopy the defects in lymphatic vessel remodeling (Bazigou et al., 2009). Among the proteins phenocopying Polydom knockout after ablation in mice, Tie1, EphrinB2, and Podoplanin are cell surface proteins expressed on LECs, of which EphrinB2 and Podoplanin bind to EphB4 and CLEC-2, respectively (Bennett et al., 1995; Christou et al., 2008; Suzuki-Inoue et al., 2007), and the ligand for Tie1 remains unknown.

Tie1 has been identified as an orphan receptor in the Angiopoietin (Ang)–Tie axis, the endothelial signaling system that governs embryonic cardiovascular and lymphatic development (Eklund et al., 2017). Tie2, another component of the Ang–Tie axis, is a receptor-type tyrosine kinase that interacts with Ang1 and Ang2, thereby modulating embryonic cardiovascular development as well as pathogenic conditions such as tumor angiogenesis and ocular- and diabetes-associated vascular diseases (Saharinen et al., 2017). Unlike Tie2, Tie1 does not bind to either Ang1 or Ang2, leaving it elusive how Tie1 exerts its function in the Ang–Tie signaling system.

Tie1-knockout or Tie1-hypomorphic mice exhibited subcutaneous edema and died immediately after birth because of breathing difficulties (Qu et al., 2010; Sato et al., 1995; Shen et al., 2014). The Tie1 mutant mice showed defects in the remodeling of the lymphatic network to form collecting vessels (Qu et al., 2010; Shen et al., 2014). Motivated by the similarities

[1]Division of Matrixome Research and Application, Institute for Protein Research, Osaka University, Suita, Japan; [2]Department of Medical Physiology, Hamamatsu University School of Medicine, Hamamatsu, Japan.

Correspondence to Kiyotoshi Sekiguchi: sekiguch@protein.osaka-u.ac.jp.



in the knockout mouse phenotypes between Polydom and Tie1, we hypothesized that Polydom is a hitherto unknown ligand for Tie1 and participates in remodeling of lymphatic vessels through its interaction with Tie1. Here, we provide evidence that Polydom directly binds to Tie1 and thereby promotes migration of LECs through activation of the PI3K/Akt signaling pathway.

## Results

### Polydom binds to Tie1 but not to Tie2

To examine whether Polydom is a ligand for Tie1, we performed solid-phase binding assays of recombinant full-length Polydom to microtiter plates coated with the extracellular domains of Tie1, Tie2, EphrinB2, and Podoplanin. EphrinB2 and Podoplanin were included because mice deficient in these proteins displayed a failure in lymphatic vessel remodeling. Integrin α9β1 was also used in the assays as a control. Consistent with a previous report (Sato-Nishiuchi et al., 2012), Polydom bound to integrin α9β1 in a divalent-cation-dependent manner (Fig. 1 A). Significant binding of Polydom was also detected for Tie1 but not for Tie2, EphrinB2, or Podoplanin. To verify the binding between Tie1 and Polydom, we performed reverse solid-phase binding assays in which Polydom was coated on the microtiter plates. Tie1 was capable of binding to Polydom in a dose-dependent manner with an apparent dissociation constant of 43 ± 4 nM (Fig. 1 B). No significant binding to Polydom was detected for Tie2 even at the highest concentration examined, thus confirming the specificity of the Tie1 binding by Polydom. These results support the hypothesis that Polydom is a hitherto unidentified ligand for Tie1 and participates in lymphatic development through its interaction with Tie1.

### Tie1 binds to the 20th CCP domain of Polydom

Next, we explored the mechanism by which Polydom binds to Tie1. Polydom was shown to be cleaved into an N-terminal 115-kD fragment (Pol-N) and a C-terminal 270-kD fragment (Pol-C) when secreted by cells (Sato-Nishiuchi et al., 2012). Solid-phase binding assays with recombinant Pol-N and Pol-C fragments (Fig. 1 C) revealed that Tie1 bound to Pol-C but not to Pol-N (Fig. 1 D), indicating that the C-terminally derived 270-kD fragment containing an array of CCP domains harbors the Tie1-binding site. To locate the Tie1-binding site within Pol-C, we produced a series of N-terminal deletion mutants of Pol-C (Fig. 1 C) and examined their ability to bind Tie1. Deletion up to the 19th CCP domain (ΔN-CCP19) did not compromise the Tie1-binding activity of pol-C, while deletion of the 20th CCP domain (ΔN-CCP20) resulted in a dramatic loss of the activity (Fig. 1 D), underscoring a critical role of the 20th CCP domain (CCP20 hereafter) for Tie1 binding. To examine whether Tie1 binds to Polydom through CCP20, we produced recombinant CCP20 as a GST fusion protein and examined its ability to bind Tie1. Tie1 bound to CCP20 but not to the 22nd CCP domain (CCP22) chosen as a control (Fig. 1 E), confirming the role of CCP20 as the Tie1-binding site in Polydom.

### Glu2567 and Gly2568 in CCP20 are required for Tie1 binding by Polydom

To further locate the amino acid residue(s) involved in Tie1 binding, we produced a series of chimeric proteins between CCP20 and CCP22 (Fig. 2 A) and examined their Tie1-binding activity. CCP20/22, in which the C-terminal half of CCP20 was swapped with that of CCP22, was capable of binding to Tie1 with an affinity comparable with that of control CCP20, while CCP22/20, in which the N-terminal half of CCP20 was swapped with that of CCP22, was inactive (Fig. 2 B), indicating that Tie1 binds to the N-terminal half of CCP20.

To narrow down the region critical for Tie1 binding, we produced a second series of chimeric CCP20 proteins, in which the N-terminal region was divided into two parts, Region-1 and Region-2, based on a sequence alignment between CCP20 and CCP22 (Fig. 2 A), and these parts were separately swapped with those of CCP22. CCP20/22-R1, a CCP20 mutant in which Region-1 was swapped with that of CCP22, was fully active in Tie1 binding, while CCP20/22-R2, a CCP20 mutant in which Region-2 was swapped with that of CCP22, was inactive (Fig. 2 B), indicating that Region-2 but not Region-1 is responsible for Tie1 binding. Consistent with this conclusion, a CCP22 mutant in which Region-2 was replaced with that of CCP20 (CCP22/20-R2) exhibited Tie1-binding activity, albeit to a lesser extent than CCP20 and CCP20/22. These results underscore the critical role of CCP20 Region-2 for Tie1 binding by Polydom.

To further determine the amino acid residues in CCP20 involved in Tie1 binding, we performed alanine-scanning mutagenesis of Region-2 in CCP20 (Fig. 2 C). Alanine substitution of Glu2567 and Gly2568 abrogated the Tie1-binding activity of CCP20, while mutations of Val2566, Ala2569, and Ile2577 resulted in partial reductions in Tie1-binding activity (Fig. 2 D). These results indicate that Glu2567 and Gly2568 are required for Tie1 binding by Polydom, although some other residues surrounding Glu2567/Gly2568 may contribute to Tie1 binding as auxiliary interaction sites. To consolidate the importance of Region-2 in CCP20 for Tie1 binding by Polydom, we produced a full-length Polydom mutant (Polydom/E2567A) in which Glu2567 in CCP20 was substituted by alanine (Fig. 2 E). The E2567A mutant was inactive in Tie1 binding (Fig. 2 F), corroborating the critical role of Glu2567 in CCP20 for Tie1 binding by Polydom.

### Polydom promotes LEC migration via Tie1

Given the similarities between the phenotypes of Polydom-null and Tie1-null mice, with both types of mice exhibiting defects in remodeling of lymphatic vessels, we examined whether the Tie1–Polydom interaction promotes LEC migration, the event that initiates lymphatic vessel remodeling. Because Polydom is secreted by the mesenchymal cells that surround LECs (Morooka et al., 2017), but not by LECs per se, we employed Transwell migration assays, in which Polydom was added to the medium in the lower chamber to attract LECs seeded in the upper chamber. As expected, LECs migrated to the lower chamber when Polydom was added, and this transmigration was not observed when other ECM proteins, such as collagen, fibronectin, and laminin, were added (Fig. 3, A and B), demonstrating the specificity of the Polydom-induced LEC migration. The Polydom-induced LEC transmigration was dose-dependent and reached a plateau at 1 µg/ml (Fig. S1). Because LEC transmigration was only observed with Polydom and not with other ECM proteins, it seems likely

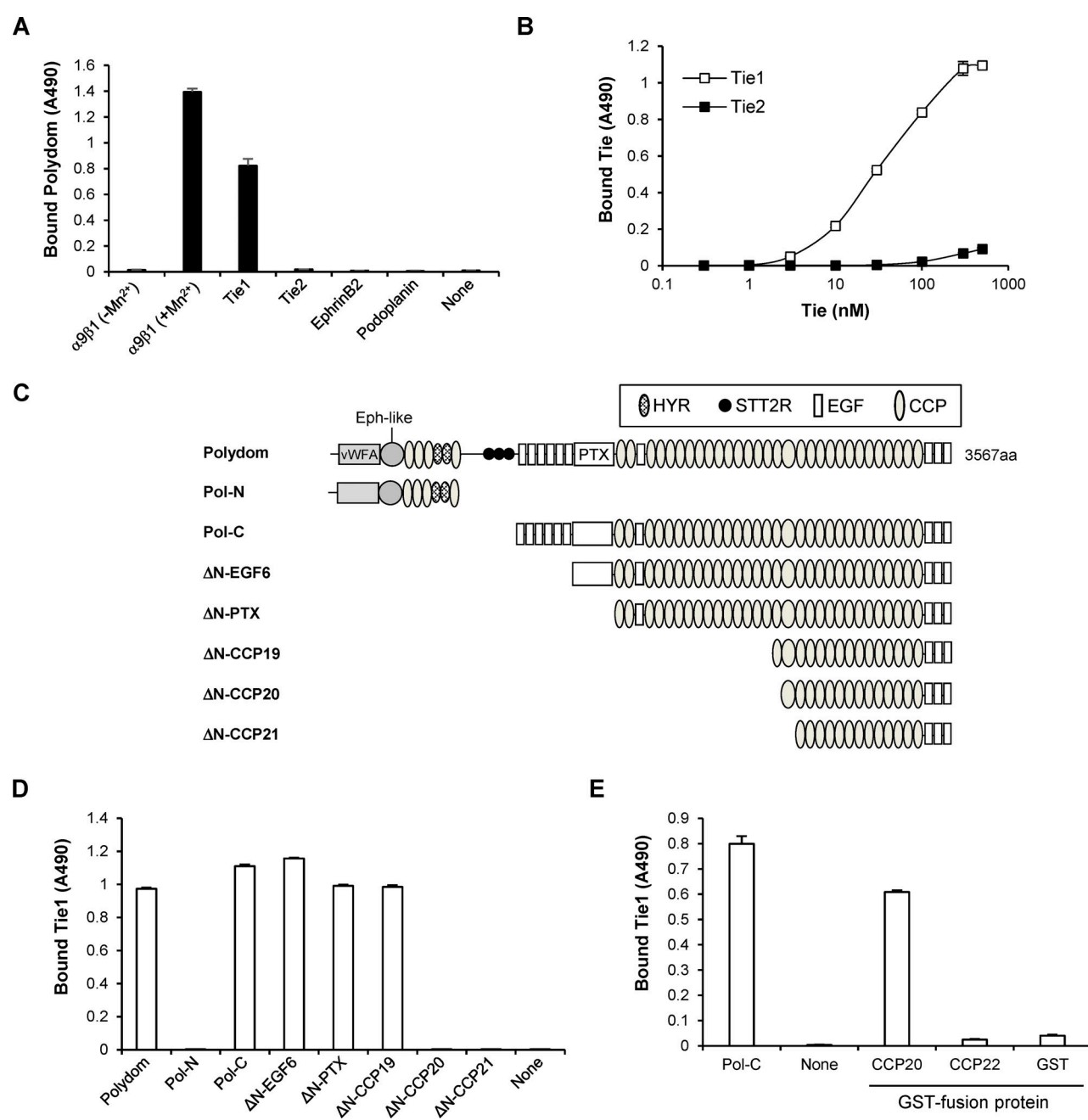

Figure 1. **Polydom binds to Tie1 via CCP20. (A)** Recombinant integrin α9β1, Tie1, Tie2, EphrinB2, and Podoplanin were coated on microtiter plates at 5 µg/ml and then incubated with full-length Polydom (5 µg/ml). Polydom was allowed to bind to integrin α9β1 in the absence or presence of 1 mM MnCl₂. The data represent means ± SD of triplicate determinations. **(B)** Titration curves of Tie1 and Tie2 binding to Polydom. Tie1 (open squares) or Tie2 (closed squares) at the indicated concentrations was allowed to bind to microtiter plates coated with full-length Polydom (10 nM). The data represent means ± SD of triplicate determinations. **(C)** Schematic representation of Polydom and its recombinant fragments. Polydom consists of multiple domains including von Willebrand factor type A (vWFA), Ephrin 2-like (Eph-like), hyaline (HYR), similar to thyroglobulin type 2 repeat (STT2R), EGF, and pentraxin (PTX) domains, and an array of CCP domains. **(D)** Polydom and its fragments were coated at 2 µg/ml on microtiter plates and incubated with Tie1 (2 µg/ml). The data represent means ± SD of triplicate determinations. **(E)** GST fusion proteins of CCP20 and CCP22 were coated at 2 µg/ml on microtiter plates and incubated with Tie1 (2 µg/ml). Pol-C and GST alone were included as positive and negative controls, respectively. The data represent means ± SD of triplicate determinations.

that Polydom promotes LEC transmigration by a chemotactic, rather than a haptotactic, mechanism. In support of this notion, Polydom was less potent than collagen, fibronectin, and laminin in promoting adhesion and subsequent spreading of LECs (Fig. S2 A). Furthermore, when the lower surface of the Transwell membrane was precoated with Polydom followed by blocking

with BSA, LEC transmigration was not observed even when Polydom was added to the medium in the lower chamber (Fig. S2, B and C). Because the lower chamber medium contains 0.5% FBS, it is conceivable that cell-adhesive protein(s) in FBS were adsorbed on the lower surface of the Transwell membrane, thereby providing adhesive substrates for LEC migration.

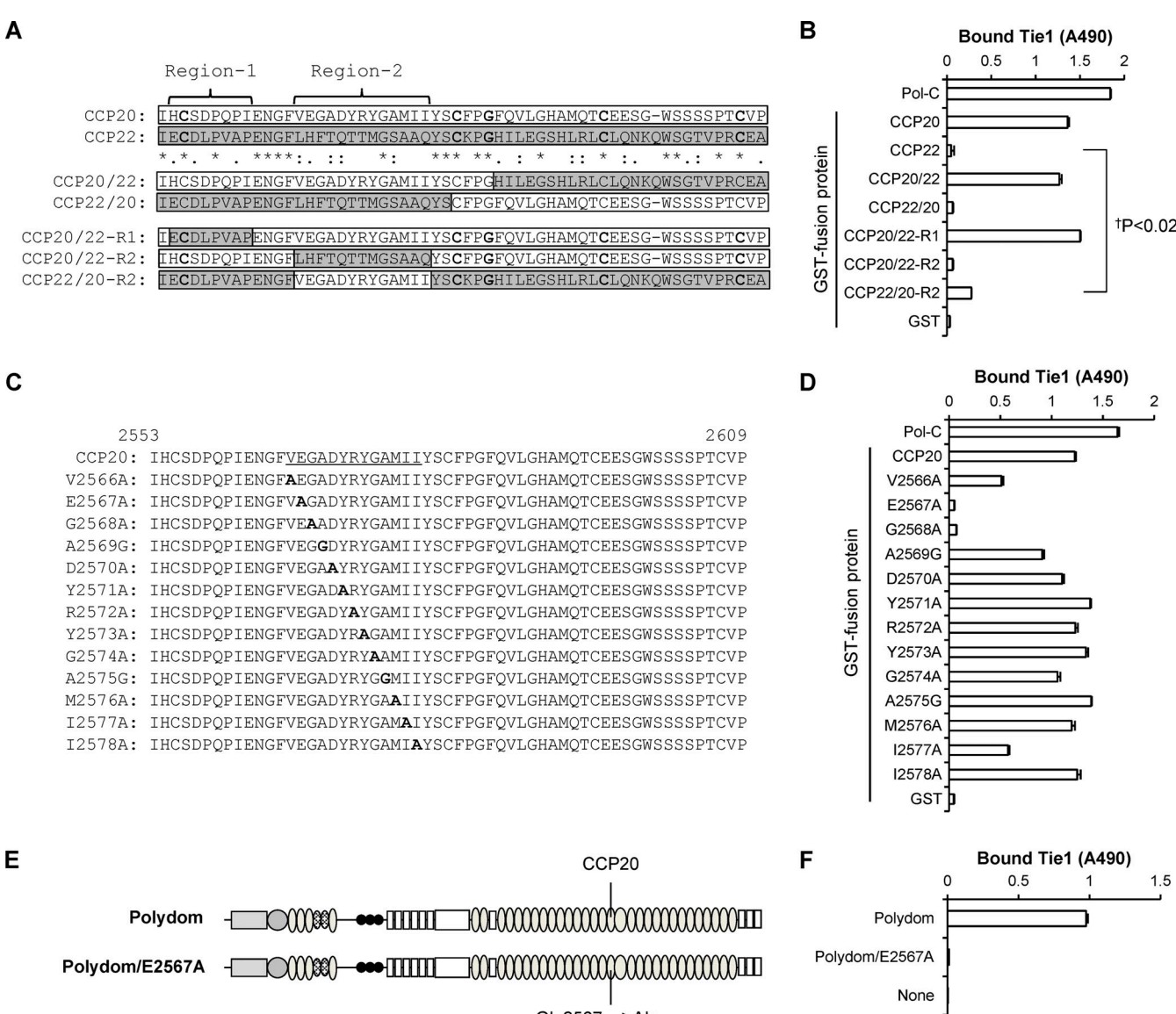

Figure 2. **Glu2567 and Gly2568 in CCP20 are required for Tie1 binding. (A)** Alignment of the amino acid sequences of CCP20 (white bars) and CCP22 (gray bars) by ClustalW. The asterisks denote identical amino acids, while the colons and periods denote strongly similar and weakly similar amino acids, respectively. The N-terminal half of CCP20/CCP22 is subdivided into Region-1 and Region-2. **(B)** Tie1 (2 µg/ml) was allowed to bind to microtiter plates coated with GST fusion proteins containing CCP20, CCP22, and their chimeras or with Pol-C or GST alone coated at 2 µg/ml. The data represent means ± SD of triplicate determinations. †P < 0.02 (n = 3). **(C and D)** A series of alanine scanning mutants of CCP20/Region-2 (C; underlined) were expressed as GST fusion proteins and assessed for their binding activities toward Tie1 (D). The data represent means ± SD of triplicate determinations. The substituted alanine (A) and glycine (G) residues are shown in bold. **(E and F)** Full-length Polydom and its Glu2567Ala mutant (Polydom/E2567A; E) were assessed for their binding activities toward Tie1 (F). The data represent means ± SD of triplicate determinations.

Consistent with this possibility, Polydom-induced LEC transmigration was abrogated by an Arg-Gly-Asp (RGD)–containing synthetic peptide but not an inactive Arg-Gly-Glu–containing synthetic peptide (Fig. S2 D). Vascular endothelial growth factor-C (VEGF-C), a lymphangiogenic factor that initiates sprouting of LECs from cardinal veins, did not promote LEC transmigration when added to the lower chamber medium (Fig. S2, E and F), suggesting that the mechanism for Polydom-induced LEC transmigration differs from that for VEGF-C–induced sprouting of LECs in the initial stage of lymphatic vessel development.

To examine whether Tie1 is involved in the Polydom-induced LEC migration, we suppressed Tie1 expression in the cells by RNA interference (Fig. 3 C). Tie2 was also knocked down as a control. Transwell migration was dramatically reduced when the cells were treated with Tie1 siRNA, while the cells treated with either control siRNA or Tie2 siRNA showed no significant reductions in transmigration (Fig. 3, D and E). The reduced LEC migration was not caused by a cytotoxic effect of Tie1 siRNA because no cleaved caspase-3 was detected in the siRNA-treated LECs (Fig. 3 F). These results indicate that Polydom promotes LEC migration through binding to Tie1.

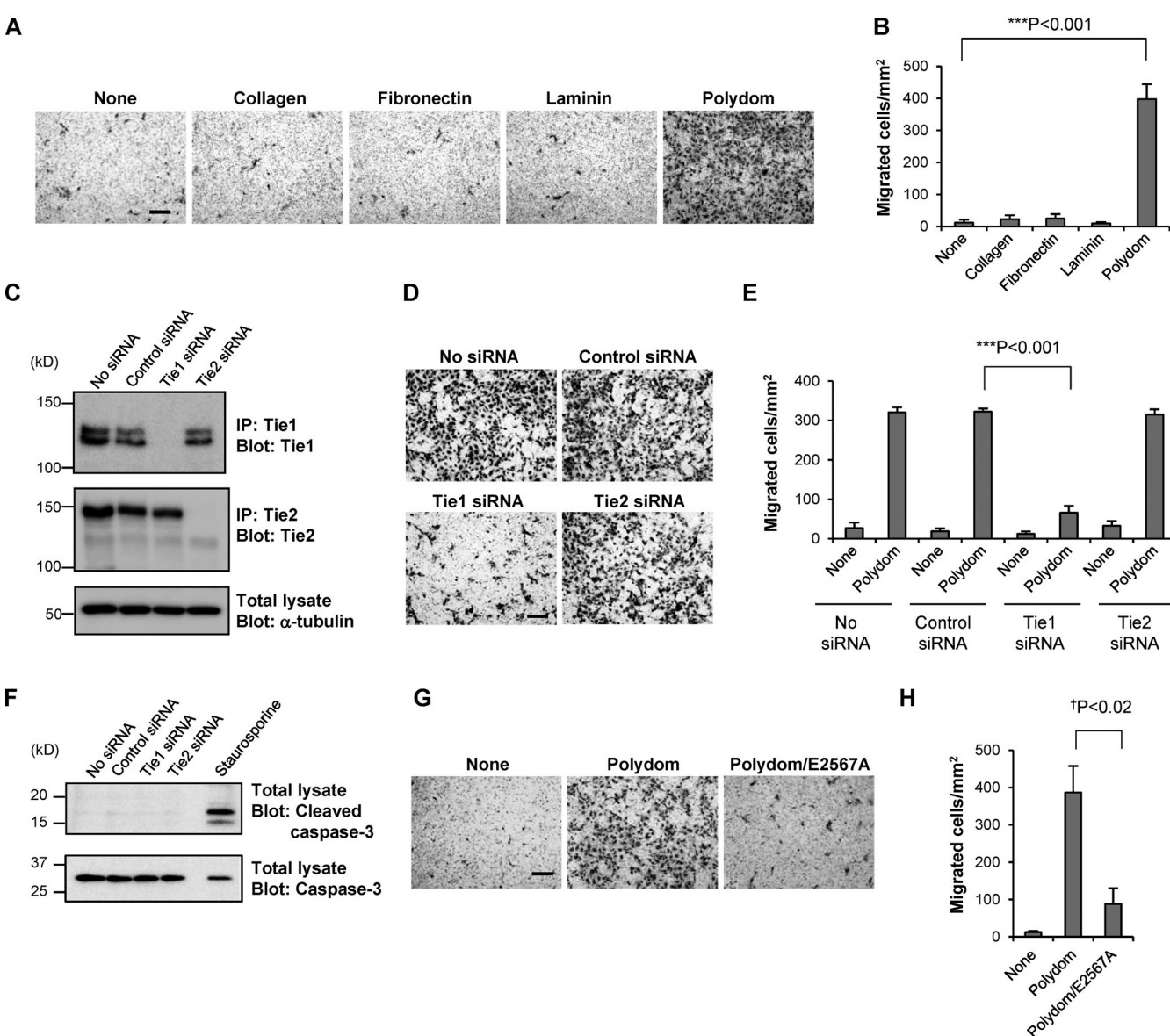

Figure 3. **Polydom promotes LEC migration via Tie1. (A)** Photographs of LECs that migrated to the lower side of Transwell membranes in the presence of collagen, fibronectin, laminin, or Polydom (3 µg/ml) added to the lower chamber medium. The molar concentration of the proteins was collagen, 23 nM; fibronectin, 15 nM; laminin, 3.8 nM; and Polydom: 8.8 nM. Bar, 200 µm. **(B)** Cells that migrated to the lower side of the membranes were counted under a microscope. The data represent means ± SD of three independent experiments each assayed in triplicate. ***P < 0.001 (n = 3). **(C)** Immunoprecipitates (IP) of Tie1 and Tie2 from lysates of LECs transfected with control, Tie1, or Tie2 siRNAs were analyzed by Western blotting under reducing conditions. The same lysates were probed with an antibody against α-tubulin as a control. The positions of molecular weight markers are shown on the left. **(D)** LECs transfected with control, Tie1, or Tie2 siRNAs were allowed to migrate in the presence of Polydom (1 µg/ml) for 16 h. Bar, 200 µm. **(E)** Cells that migrated to the lower side of the membranes were counted under a microscope in the absence (none) or presence of Polydom added to the lower chamber medium. The data represent means ± SD of three independent experiments each assayed in triplicate. ***P < 0.001 (n = 3). **(F)** Total lysates of LECs transfected with control, Tie1, or Tie2 siRNAs were analyzed by Western blotting using anti-cleaved caspase-3 and anti-caspase-3 antibodies under reducing conditions. Lysates of LECs treated with staurosporine (0.2 µM, 4 h) were used as a positive control for cleaved caspase-3 detection. **(G)** Photographs of LECs that migrated to the lower side of the membranes in the presence of Polydom or Polydom/E2567A (1 µg/ml) in the lower chamber medium. Bar, 200 µm. **(H)** Cells that migrated to the lower side of the membranes were counted under a microscope. The data represent means ± SD of three independent experiments each assayed in triplicate. †P < 0.02 (n = 3). Source data are available for this figure: SourceData F3.

To corroborate the involvement of Tie1 in Polydom-induced LEC migration, we performed Transwell migration assays with Polydom/E2567A, a full-length Polydom mutant lacking Tie1-binding activity. When Polydom/E2567A was added to the lower chamber, LEC transmigration was significantly compromised, exhibiting an ∼80% reduction compared with that induced by control Polydom (Fig. 3, G and H). These results demonstrated that Polydom functions as a Tie1 ligand and promotes LEC migration through binding to Tie1.

**Polydom activates the PI3K–Akt signaling pathway in LECs**

We further sought to identify the signaling pathways involved in the Polydom-induced LEC migration. Because Tie1 is a receptor tyrosine kinase and associates with Tie2, we explored whether Polydom could promote tyrosine phosphorylation of Tie1. However, we did not detect any significant phosphorylation of Tie1 or Tie2 in LECs treated with Polydom (Fig. S3 A). We also examined whether Polydom could induce Tie1 phosphorylation in 293-F cells transfected with recombinant Tie1 and/or Tie2. However, no significant Tie1 phosphorylation was induced by Polydom in Tie1-transfected 293-F cells, either (Fig. S3 B). Notably, Tie1 phosphorylation was induced in 293-F cells cotransfected with Tie1 and Tie2 irrespective of the presence or absence of Polydom, consistent with previous observations that Tie1 was only phosphorylated when Tie1 and Tie2 were coexpressed in 293 cells (Korhonen et al., 2016; Saharinen et al., 2005; Yuan et al., 2007).

Because the PI3K/Akt pathway is known to be activated downstream of the Ang–Tie signaling system (Kim et al., 2000; Kontos et al., 1998; Korhonen et al., 2016), we investigated the involvement of the PI3K/Akt pathway in the Polydom-induced LEC migration using small-molecule inhibitors. LY294002 and Wortmannin, specific inhibitors of PI3K activity, inhibited the Polydom-induced transmigration of LECs, while SCH772984, an inhibitor of extracellular signal-regulated kinase (ERK), had no inhibitory effect on LEC migration (Fig. 4 A). Supporting the involvement of the PI3K/Akt pathway in Polydom-induced LEC migration, Akt phosphorylation was enhanced in Polydom-treated LECs (Fig. 4 B). The Polydom/E2567A mutant, which is inactive for Tie1 binding, did not induce Akt phosphorylation, further corroborating the involvement of the Polydom–Tie1 interaction in activation of the PI3K/Akt pathway.

One of the signaling events downstream of the PI3K/Akt pathway is Forkhead box protein O1 (FOXO1) inactivation and nuclear exclusion (Daly et al., 2004). FOXO1 nuclear exclusion was found to be attenuated in Tie1-silenced vascular endothelial cells (Korhonen et al., 2016). To examine whether Polydom inactivates FOXO1 by nuclear exclusion, we immunostained LECs for FOXO1 with and without Polydom treatment. Nuclear exclusion of FOXO1 was induced in LECs upon Polydom treatment, while FOXO1 remained in the nucleus in untreated LECs (Fig. 4, C and D), consistent with the increased Akt phosphorylation in Polydom-treated LECs. We further examined whether Polydom could regulate the activation/inactivation status of Foxo1 in vivo by Foxo1 immunostaining of lymphatic vessels in wild-type and Polydom-deficient mice. Foxo1 was exclusively detected in the nucleus of LECs in Polydom-deficient mice, while it was detected in both the nucleus and cytoplasm of LECs in wild-type mice (Fig. 4, E and F; and Fig. S4). The nuclear Foxo1 signals were more pronounced in the Polydom-deficient mice compared with the wild-type mice. These results indicate that Polydom binds to Tie1 and facilitates lymphatic vessel remodeling by modulating the PI3K/Akt signaling pathway in LECs.

## Discussion

Tie1 has long been known as an orphan receptor in the Ang–Tie axis. Unlike Tie2, which serves as the primary receptor in the axis and binds to both Ang1 and Ang2, the function of Tie1 has remained elusive due to the absence of a cognate ligand. None of the known angiopoietins bind to Tie1 (Jones et al., 2001). Here, we provide evidence that Polydom binds to Tie1 but not to Tie2 and promotes LEC migration in a Tie1-dependent manner. Polydom binds to Tie1 through CCP20 with an apparent dissociation constant of 43 nM and requires Glu2567 and Gly2568 within CCP20 for Tie1 binding. Polydom promotes LEC migration in Transwell assays, which is abrogated by siRNA interference of Tie1 but not of Tie2. The critical role of the Polydom–Tie1 interaction for LEC migration was confirmed by abrogation of the Polydom-induced LEC migration by the Glu2567Ala mutation in CCP20. Consistent with the role of the Polydom–Tie1 interaction in LEC migration, Tie1-deficient or Tie1-hypomorphic mice showed defects in remodeling of the lymphatic plexus into collecting vessels (Qu et al., 2010; Shen et al., 2014), similar to the observations in Polydom-deficient mice (Karpanen et al., 2017; Morooka et al., 2017). It should be noted that Tie1 expression in LECs is significantly reduced in Polydom-deficient mice (Morooka et al., 2017). Because Polydom is deposited around lymphatic vessels (Morooka et al., 2017), Tie1 ligation by Polydom may stabilize the Tie1 expression on LECs.

Although Tie1 is a receptor tyrosine kinase with a functionally active kinase domain (Kontos et al., 2002), it remains unclear whether Tie1 is phosphorylated under physiological conditions. There is accumulating evidence that Tie1 forms a heterocomplex with Tie2 and becomes phosphorylated in a Tie2-dependent manner upon stimulation with Ang1, while it is barely phosphorylated in the absence of Tie2 (Korhonen et al., 2016; Saharinen et al., 2005; Yuan et al., 2007). Consistent with these findings, we found that Tie1 phosphorylation was only induced in 293-F cells when Tie1 was co-transfected with Tie2. Because Polydom does not enhance Tie1 phosphorylation in either LECs or 293-F cells transfected with Tie1, it is conceivable that the signaling events downstream of the Polydom–Tie1 interaction are not linked to Tie1 or Tie2 phosphorylation. Nevertheless, even in the absence of stimulation by Tie1 or Tie2 phosphorylation, our results showed that Polydom activated the PI3K/Akt signaling pathway, as evidenced by the enhanced Akt phosphorylation and subsequent FOXO1 nuclear exclusion, a hallmark signaling event downstream of Ang–Tie interactions (Daly et al., 2004), in Polydom-treated LECs. The involvement of Polydom in activation of the PI3K/Akt pathway was further corroborated by the in vivo observation that Foxo1 nuclear exclusion was significantly compromised in Polydom-deficient mice. The mechanism by which the Polydom–Tie1 interaction activates the PI3K/Akt pathway without Tie1/Tie2 phosphorylation remains to be explored. Because Tie1 and Tie2 were reported to associate with integrins α5β1 and αvβ3 (Cascone et al., 2005; Dalton et al., 2016) and thereby modulate the PI3K/Akt pathway in an integrin-dependent manner (Dalton et al., 2016), it is tempting to speculate that these integrins are also involved in the activation of the PI3K/Akt signaling pathway downstream of the Polydom–Tie1 interaction. Essential roles of integrins in activating intracellular signaling cascades, including the PI3K/Akt pathway, are well established (Cooper and Giancotti, 2019; Danen and

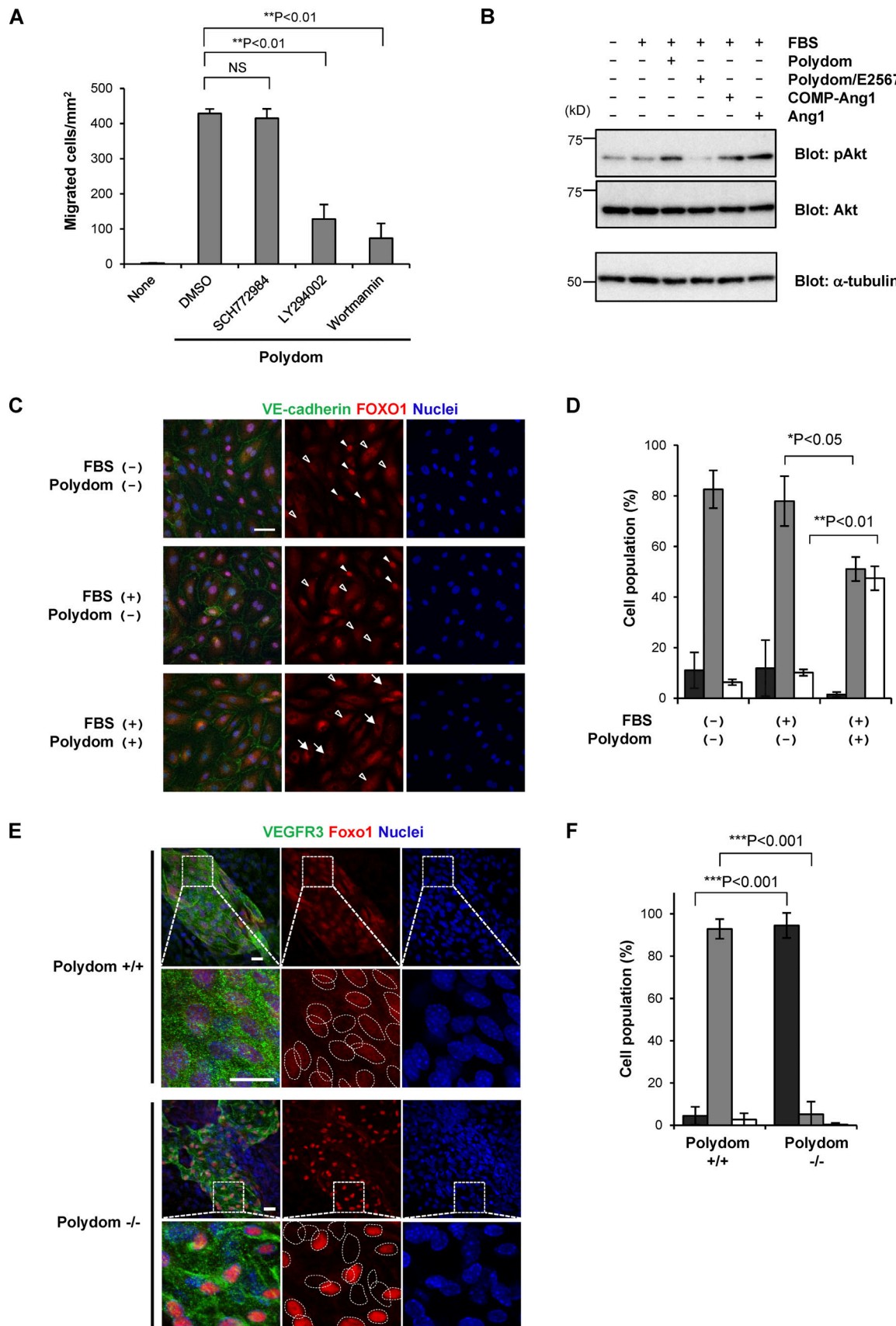

Figure 4. **Involvement of the PI3K/Akt signaling pathway in Polydom-induced LEC migration. (A)** LECs were allowed to migrate to the lower side of Transwell membranes in the absence (none) or presence of Polydom (1 μg/ml) with an ERK inhibitor (2 μM SCH772984) or PI3K inhibitors (20 μM LY294002 or

2 µM Wortmannin) added to the lower chamber medium. Cells that migrated to the lower side of the membranes were counted under a microscope. The data represent means ± SD of three independent experiments each assayed in triplicate. NS, not significant; **P < 0.01 (n = 3). **(B)** Serum-starved LECs were treated with Polydom (1 µg/ml), Polydom/E2567A (1 µg/ml), COMP-Ang1 (500 ng/ml), or Ang1 (500 ng/ml) in medium containing 0.5% FBS at 37°C for 15 min. Total lysates of the cells were immunoblotted for pAkt or total Akt under reducing conditions. **(C)** Serum-starved LECs were treated with Polydom (1 µg/ml) in medium with or without 0.5% FBS at 37°C for 30 min, followed by immunostaining for VE-cadherin (green) and FOXO1 (red). Nuclei were stained with Hoechst 33342 (blue). Cells in which FOXO1 was excluded from the nucleus (arrows) were frequently observed in the presence of Polydom, while FOXO1 was detected exclusively in the nucleus (closed arrowheads) or both the nucleus and cytoplasm with reduced signal intensity (open arrowheads) in the absence of Polydom. Bar, 20 µm. **(D)** Quantification of the percentages of cells showing FOXO1 localization in the nucleus only (black), in both the nucleus and the cytoplasm (gray), and in the cytoplasm only (white). The data represent means ± SD of three independent experiments. *P < 0.05 and **P < 0.01 (n = 3). **(E)** Whole-mount immunofluorescence staining of the skin of E17.5 wild-type (Polydom+/+, upper panels) and Polydom-deficient mice (Polydom−/−, lower panels) for VEGFR3 (green, used as a lymphatic vessel marker) and Foxo1 (red). Nuclei were stained with Hoechst 33342 (blue). Magnified views of the boxed areas are shown below. Individual nuclei are encircled with dotted lines in the magnified views for Foxo1 staining. Bars, 20 µm. **(F)** Quantification of the percentages of VEGFR3-positive cells showing Foxo1 localization in the nucleus only (black), both in the nucleus and the cytoplasm (gray), and in the cytoplasm only (white) in Polydom+/+ and Polydom−/− mice. The data represent means ± SD. ***P < 0.001 (n = 3 per genotype). Source data are available for this figure: SourceData F4.

Yamada, 2001). In support of this notion, Polydom-induced LEC transmigration was abrogated by an RGD-containing synthetic peptide, indicating the involvement of RGD-binding integrins, including integrins α5β1 and αvβ3. Expression of integrins α5β1 and αvβ3 by the LECs used in this study was previously confirmed by flow cytometric analysis (Garmy-Susini et al., 2007). Therefore, it seems likely that FBS-derived RGD-containing adhesive protein(s), such as fibronectin, were adsorbed onto the Transwell membrane and recognized by the RGD-binding integrins on LECs, thereby sensitizing the signaling pathway downstream of the Polydom–Tie1 interaction in our Transwell migration assay. It remains to be elucidated which integrin(s) cooperate with the Ang–Tie signaling system and how they potentiate the signaling pathway downstream of the Polydom–Tie1 interaction without enhanced Tie1/Tie2 phosphorylation.

Polydom was previously shown to bind to integrin α9β1, which is expressed in lymphatic valves (Bazigou et al., 2009). Depletion of integrin α9 in mouse embryos resulted in failure of lymphatic valve development, although remodeling of the lymphatic plexus into collecting vessels was not impaired (Bazigou et al., 2009). Thus, the interaction of Polydom with integrin α9β1 is not considered instrumental for lymphatic vessel remodeling. Consistent with this conclusion, knockdown of integrin α9 did not compromise LEC migration in our Transwell assay (Fig. S5, A–C). Furthermore, a full-length Polydom mutant in which the integrin α9β1 recognition sequence ED**DMME**VPY was substituted by ED**AMMA**VPY (Sato-Nishiuchi et al., 2012) was fully active in promoting LEC transmigration, even though it was inactive in binding to integrin α9β1 (Fig. S5, D–F). It should also be noted that depletion of Polydom after embryonic day 15.5 did not compromise the formation of collecting lymphatic vessels but did result in impaired luminal valve formation (Morooka et al., 2017), indicating that Polydom is involved in not only lymphatic vessel remodeling but also lymphatic valve formation. Furthermore, postnatal depletion of Tie1 resulted in impaired maturation of collecting lymphatic vessels as well as impaired valve formation (Shen et al., 2014). These observations suggest that the interactions between Polydom–Tie1 and Polydom–integrin α9β1 may cooperatively contribute to lymphatic valve formation following remodeling of the lymphatic plexus into collecting vessels.

In addition to Tie1 and integrin α9β1, Polydom binds to Ang1 and Ang2 (Morooka et al., 2017). Ang2 deficiency in mice leads to failure of lymphatic vessel remodeling (Dellinger et al., 2008; Gale et al., 2002), a phenotype mimicking that of Polydom-knockout mice, raising the possibility that Polydom may also contribute to remodeling of lymphatic vessels through its interaction with Ang2. Ang2 binds to Tie2 with a comparable affinity to that of Ang1, but its role as a Tie2 agonist remains controversial. Thus, Ang2 is capable of activating Tie2 as an agonist at high concentrations but behaves as an antagonist that prevents Tie2 activation under physiological conditions (Maisonpierre et al., 1997). Because Transwell cell migration was not compromised by Tie2 knockdown, it seems unlikely that Ang2 functions as a Tie2 agonist in Polydom-induced LEC migration. Because Ang2 is secreted by LECs as a homodimer and/or oligomer (Davis et al., 2003; Jang et al., 2009; Veikkola et al., 2003) and is capable of binding to Polydom, it is tempting to speculate that Ang2 operates as a crosslinker of Polydom monomers and increases the avidity of the Polydom–Tie1 interaction, thereby enabling the signaling events elicited downstream of the interaction, including the activation of the PI3K/Akt signaling pathway. Further investigation of the mechanistic basis of the Polydom–Ang2 interaction will reconcile the apparently controversial functions of Ang2 in the Ang–Tie signaling system under physiological and pathological conditions.

Our results showed that the Glu2567 and Gly2568 residues in CCP20 were required for Tie1 binding by Polydom. These residues in CCP20 are conserved in not only mammals but also other vertebrates including chicken, *Xenopus*, and zebrafish (Fig. 5). This is in sharp contrast to the integrin α9β1 recognition site in Polydom, i.e., the EDDMMEVPY sequence in CCP21, which is only conserved in mammals (Sato-Nishiuchi et al., 2012). Consistent with the conserved Tie1-binding sequence in vertebrates, lymphatic development was severely impaired in zebrafish after knockdown/knockout of the Polydom gene (Karpanen et al., 2017; Morooka et al., 2017). Homozygous Polydom mutant zebrafish developed severe edema and lacked the thoracic duct, an equivalent to cardinal lymphatic vessels in mammals. Tie1 knockout in zebrafish was also shown to cause severe edema (Carlantoni et al., 2021). Recently, Hußmann et al. reported that multiple phenotypic traits in the lymphatic and blood vasculature were shared between *tie1*-null and *svep1/polydom*-null zebrafish, and proposed a genetic interaction between svep1/polydom and tie1 (Hußmann et al., 2022 Preprint). It should be noted that defects in venous development were observed in the

```
                              2567 2568
   Mouse_CCP20   IHCSDPQPIENGFVEGADYRYGAMIIYSCFPGFQVLGHAMQTCEESGWSSSSPTCVP
   Human_CCP20   IHCDSPQPIENGFVEGADYSYGAIIIYSCFPGFQVAGHAMQTCEESGWSSSIPTCMP
  Rabbit_CCP20   IHCNPPQPIENGFVEGADYSYGAVIIYSCFPGFQVAGHAMQTCKESGWSSSIPTCVP
 Chicken_CCP20   INCQPPQPIENGFVEGADYSYGAMVIYSCVPGFQLSGLAMQTCEESGWSSSTPTCLP
 Xenopus_CCP20   ITCDPPQPIENGFVEGADYSYGAMIIFSCMPGFQLVGLAMQTCEESGWSSSTPVCLH
Zebrafish_CCP20   IYCSPPKPIDNGFVEGRDRKFGVTIFYSCFPGFLLVGNNHLTCEDHGWSSSEPKCVL
                  * *. *: *:****** *  :*. ::::**.*** : *   **:: ***** * *:
```

Figure 5. **Multiple sequence alignment of CCP20 domains.** The amino acid sequences of the CCP20 domains of mouse, human, rabbit, chicken, *Xenopus*, and zebrafish Polydom were aligned by ClustalW. The asterisks denote identical amino acids, while the colons and periods denote strongly similar and weakly similar amino acids, respectively. The glutamate (E) and glycine (G) residues corresponding to E2567 and G2568 in mouse Polydom (highlighted in pink), which are critically required for Tie1 binding, are conserved in these animals.

skin of Polydom-deficient mice but were not observed in the cardinal vein or mesenteries (Morooka et al., 2017). Defects in venous formation in the skin but not in the cardinal vein were also reported for Tie1-deficient mice (Cao et al., 2022 *Preprint*), suggesting that the Polydom–Tie1 interaction is not only involved in lymphangiogenesis but also at least partially involved in venogenesis in mice, as demonstrated for *svep1/polydom*-knockout and *tie1*-knockout zebrafish. Taken together, these findings support the conclusion that Polydom is a physiological ligand for Tie1 and modulates the Ang–Tie signaling system through its interactions with Tie1 and possibly Ang1/Ang2, thereby ensuring lymphatic vessel remodeling.

In summary, we have provided evidence that Polydom binds to Tie1 and activates the PI3K/Akt signaling pathway without promoting Tie1/Tie2 phosphorylation. Polydom promotes LEC migration through binding to Tie1 in a PI3K-dependent manner. Given that Polydom-knockout mice and zebrafish exhibit defects in lymphatic vessel development and develop severe lymphedema, resembling the phenotypes of Tie1-knockout mice and zebrafish, these findings lead us to conclude that Polydom is a hitherto unidentified ligand for Tie1 that plays an important role in lymphatic vessel development through modulation of the Ang–Tie signaling system. Because Polydom binds to not only Tie1 but also integrin α9β1, Ang1, and Ang2, all of which are involved in lymphatic and/or blood vessel development, further investigation of how Polydom interacts with these lymphangiogenic components and orchestrates the signaling pathways downstream of these components will contribute to a better understanding of the molecular basis for lymphangiogenesis in development and human diseases.

## Materials and methods

### Antibodies and reagents
Goat polyclonal antibodies against human Tie1 (AF619), human Tie2 (AF313), and mouse vascular endothelial growth factor receptor 3 (VEGFR3; AF743) were purchased from R&D Systems. Rabbit anti-human Tie1 (C-18; cross-reacts with Tie2 to a lesser extent), mouse anti-phosphotyrosine (4G10), mouse anti-α-tubulin (2F9), and mouse anti-human vascular endothelial cadherin (VE-cadherin; BV9) antibodies were obtained from Santa Cruz Biotechnology, GeneTex, MBL, and BioLegend, respectively. A mAb against integrin α9β1 (Y9A2) was purchased from Millipore. Mouse IgG isotype control (#02-6502) was obtained from Zymed Laboratories. Rabbit antibodies against phospho-

Akt Ser473 (#9271), Akt (#9272), cleaved caspase-3 Asp175 (5A1E, #9664), caspase-3 (#9662), and FOXO1 (C29H4, #2880) were purchased from Cell Signaling Technology. HRP-conjugated donkey anti-rabbit IgG (#711-035-152), anti-human IgG (#709-035-149), anti-mouse IgG (#715-035-150), and anti-goat IgG (#705-035-147) were obtained from Jackson ImmunoResearch. Alexa Fluor 488–conjugated goat anti-mouse IgG (H+L; #A11029) and donkey anti-goat IgG (H+L; #A11055) and Alexa Fluor 546–conjugated donkey anti-rabbit IgG (H+L; #A10040) were purchased from Invitrogen. Recombinant Tie1 and Tie2, both containing their extracellular domains and expressed as Fc fusion proteins, recombinant human VEGF-C, and human Ang1 were obtained from R&D Systems. Rabbit antibodies against Pol-N and recombinant integrin α9β1 were prepared as described (Sato-Nishiuchi et al., 2012). Type I collagen (Cellmatrix type I-A) was purchased from Nitta Gelatin. Plasma fibronectin was purified from outdated human plasma by gelatin-affinity chromatography (Sekiguchi et al., 1983). Mouse laminin-111 was purified from mouse Engelbreth–Holm–Swarm tumor tissues, as previously described (Paulsson et al., 1987). Staurosporine was obtained from Adipogen. Inhibitors for ERK (SCH772984) and PI3K (LY294002 and Wortmannin) were purchased from Selleck Biotech. Synthetic GRGDSP and GRGESP peptides were purchased from Greiner Bio-One.

### Construction of expression vectors
The expression vector for mouse Polydom with an N-terminal FLAG tag and a C-terminal His$_6$ tag was constructed as described previously (Sato-Nishiuchi et al., 2012). Expression vectors for truncated forms of Polydom (Pol-N, Pol-C, ΔN-EGF6, ΔN-PTX, ΔN-CCP20, ΔN-CCP21) with a C-terminal His$_6$ tag were constructed as described (Sato-Nishiuchi et al., 2012). Expression vectors for ΔN-CCP19, a truncated form of Polydom with N-terminal deletion up to the 19th CCP domain and a C-terminal His$_6$ tag, full-length Polydom E2567A mutant (Polydom/E2567A), and full-length Polydom D2638A/E2641A mutant (Polydom/AMMA) with an N-terminal FLAG tag and a C-terminal His$_6$ tag were constructed in this study. An expression vector for the Fc region of human immunoglobulin G1 (hIgG1-Fc) was provided by Dr. Mitsuharu Hattori (Graduate School of Pharmaceutical Sciences, Nagoya City University, Nagoya, Japan). A DNA segment encoding hIgG1-Fc and a FLAG tag was amplified by PCR and inserted to the NotI and ApaI sites of pSecTag2B (Thermo Fisher Scientific), yielding pSecTag2B-Fc-FLAG. A cDNA segment encoding the extracellular region of

human EphrinB2 (amino acids 28–229) was obtained by RT-PCR using RNA extracted from human dermal LECs (PromoCell). A cDNA segment encoding the extracellular region of human Podoplanin (amino acids 23–131) was amplified by PCR using MGC clone 4876446 (Open Biosystems) as a template and inserted into pSecTag2B-Fc-FLAG at the HindIII/NotI sites. cDNA segments encoding full-length human Tie1 (amino acids 1–1138) and Tie2 (amino acids 1–1124) were obtained by RT-PCR using RNA extracted from human LECs and inserted with a cDNA segment encoding C-terminal FLAG tag (DYKDDDDK) or HA tag (YPYDVPDYA), respectively, into the pcDNA3.1 vector at the KpnI/ApaI sites. The expression vector for COMP-Ang1 was constructed as follows: a cDNA segment encoding human Ang1 linker and fibrinogen-like domain (amino acids 255–498) was amplified by PCR using human cDNA clone SC111587 (OriGene Technologies) as a template and inserted into pSecTag2B at the HindIII/ApaI sites with the cDNA segment encoding an N-terminal FLAG tag and a short coiled-coil domain of human cartilage oligomeric matrix protein (COMP; amino acids 28–73, amplified by PCR using synthetic primers; Cho et al., 2004).

### Expression and purification of recombinant proteins
Recombinant Polydom, Polydom fragments, FLAG-tagged Fc-fusion proteins, and FLAG-tagged COMP-Ang1 were produced using a Freestyle 293 Expression System (Thermo Fisher Scientific). Freestyle 293 cells were transfected with expression vectors using 293fectin (Thermo Fisher Scientific) and grown in serum-free FreeStyle 293 expression medium for 72 h. The conditioned media were collected and clarified by centrifugation. For purification of FLAG-tagged proteins, conditioned media were applied to a DDDDK-tagged protein purification gel (MBL) column and bound proteins were eluted with 100 µg/ml DDDDK-tagged peptide in TBS. For purification of His-tagged proteins, conditioned media were subjected to affinity chromatography using cOmplete His-tag purification resin (Roche). The columns were washed with TBS containing 5 mM imidazole and the bound proteins were eluted with TBS containing 250 mM imidazole. The eluted proteins were dialyzed against PBS. The concentrations of the purified proteins were determined by the Bradford assay using BSA as a standard.

### Expression and purification of GST-fused Polydom fragments
cDNAs encoding CCP20, CCP22, or their chimeras were amplified by PCR and subcloned into pGEX4T-1 (GE Healthcare) at the EcoRI/SalI sites. GST fusion proteins were induced in *Escherichia coli* BL21 cells by incubation with 0.1 mM IPTG at 25°C for 2 h. The cells were lysed by sonication and the supernatants were passed over a glutathione-Sepharose 4B (GE Healthcare) column. The bound proteins were eluted with 50 mM Tris-HCl (pH 8.0) containing 10 mM glutathione. The eluted proteins were dialyzed against 20 mM Hepes buffer (pH 8.0) containing 130 mM NaCl. The concentrations of the purified proteins were determined by the Bradford assay.

### Solid-phase binding assays
For Polydom binding assays, microtiter plates (Maxisorp; Nunc) were coated with 5 µg/ml EphrinB2, Podoplanin, Tie1, Tie2, or integrin α9β1 overnight at 4°C and then blocked with TBS containing 1% BSA for 1 h at room temperature. After washing with TBS containing 0.1% BSA and 0.02% Tween-20 (Buffer W), Polydom (5 µg/ml) was allowed to bind to the microtiter plates by incubation for 2 h at room temperature. After three washes with Buffer W, bound Polydom was quantified by incubation with an anti-pol-N antibody for 1 h and an HRP-conjugated anti-rabbit IgG antibody for 40 min, followed by incubation with o-phenylenediamine for 10 min and measurement of the absorbance at 490 nm. In reverse solid-phase assays, microtiter plates were coated with 2 µg/ml Polydom, Polydom deletion mutants, or GST-fused Polydom fragments, and then incubated with serially diluted Tie1 or Tie2 with an Fc tag, followed by quantification of bound Tie1 or Tie2 using an HRP-conjugated anti-human IgG antibody against the Fc tag. The apparent dissociation constant was determined as described previously (Nishiuchi et al., 2003). Integrin binding assays were performed as described previously (Sato-Nishiuchi et al., 2012). The assays were performed in triplicate and the results were confirmed by three independent experiments.

### Transwell migration assays
Cell migration was evaluated using a 24-well Transwell chamber with 8-µm pore inserts (Corning Costar). To precoat the lower side of the Transwell chamber, a 100-µl aliquot of Polydom (2 µg/ml) was dropped on the Transwell chamber placed upside down and incubated overnight at 4°C. After removal of the coating solution by aspiration, the lower side of the Transwell chamber was blocked with 1% BSA for 1 h at 37°C, followed by washing with PBS. Human dermal LECs (PromoCell) were maintained in Endothelial Cell Growth Medium (EGM)-MV2 (PromoCell) containing EGF (5 ng/ml), basic fibroblast growth factor (10 ng/ml), IGF (20 ng/ml), VEGF-A-165 (0.5 ng/ml), ascorbic acid (1 µg/ml), and hydrocortisone (0.2 µg/ml), and grown at 37°C in a humidified atmosphere containing 5% $CO_2$. The cells were starved in Endothelial Cell Basal Medium (EBM)-MV2 (PromoCell) overnight, detached, and resuspended in EBM-MV2 containing 0.1% BSA to a density of $5 \times 10^5$ cells/ml. Next, 100-µl aliquots of cells were seeded in each upper chamber, and EBM-MV2 containing 0.5% FBS and Polydom (or other effector proteins) at specified concentrations was added into the lower chamber. Inhibitors for ERK (SCH772984; 2 µM) or PI3K (LY294002, 20 µM; Wortmannin, 2 µM), as well as GRGDSP and GRGESP peptides (200 µM), were added to the lower chamber medium. After incubation at 37°C in a humidified atmosphere containing 5% $CO_2$ for 16 h, cells that remained in the upper chamber were removed with a cotton swab. The migrated cells were fixed with 3.7% formaldehyde and stained with 0.1% toluidine blue. The migrated cells were imaged using an inverted microscope (CK40; Olympus) with a SPlan4PL 4×/0.13 objective lens (Olympus) and photographed with an AxioCam ERc 5s camera (Zeiss) and AxioVision software (Zeiss). The cell number was counted using ImageJ software (http://rsb.info.nih.gov/ij/). The assays were performed in triplicate and the results were confirmed by three independent experiments.

## RNA interference

siRNA mixtures targeting human Tie1 (siGENOME Human Tie1 siRNA SMARTpool, M-003179-02), human Tie2 (siGENOME Human TEK siRNA SMARTpool, M-003178-03), or human integrin α9 (siGENOME Human ITGA9 siRNA SMARTpool, M-008005-01) and a negative control mixture (siGENOME Non-Targeting siRNA Pool #2, D-001206-14) were purchased from Dharmacon/Horizon Discovery. LECs were grown on 10-cm dishes in EGM-MV2 and transfected with 10 pmol (for Tie1 and Tie2) or 20 pmol (for ITGA9) of siRNA using Lipofectamine RNAiMAX (Thermo Fisher Scientific). The medium was replaced with fresh EGM-MV2 at 5 h after transfection and then replaced with EBM-MV2 on the following day. After overnight starvation in EBM-MV2, the cells were used for Transwell cell migration assays. The cells were also subjected to immunoprecipitation or flow cytometry 3 d after transfection.

## Cell adhesion assay

Cell adhesion assays were performed as described previously (Sato et al., 2009). Briefly, microtiter plates were coated overnight at 4°C with substrate proteins. The plates were blocked with blocking medium (EBM-MV2 containing 1% BSA) for 1 h at 37°C. LECs resuspended in blocking medium were plated at 4.0 × $10^4$ cells/well and incubated for 30 min at 37°C in a humidified atmosphere containing 5% $CO_2$. After removal of non-adherent cells by washing with blocking medium, the attached cells were fixed with 3.7% formaldehyde, washed three times with PBS, and stained with 0.1% toluidine blue in PBS. The adhered cells were imaged using an inverted microscope (CK40; Olympus) with a SPlan4PL 4×/0.13 objective lens (Olympus) and photographed with an AxioCam ERc 5s camera (Zeiss) and AxioVision software (Zeiss).

## Immunoprecipitation and Western blotting

After overnight serum starvation, the culture medium of LECs was replaced with EBM-MV2 containing 0.5% FBS, 1 μg/ml Polydom (or Polydom/E2567A), or 500 ng/ml Ang1 (or COMP-Ang1), and incubated for 15 min at 37°C in a humidified atmosphere containing 5% $CO_2$. After washing with washing buffer (10 mM Tris-HCl, pH 7.4, 0.25 M sucrose), the cells were homogenized in lysis buffer (50 mM Tris-HCl, pH 7.5, 150 mM NaCl, 0.2% Triton X-100, 0.5 mM EDTA, 1 mM PMSF, 10 mM NaF, 10 μM $Na_2MoO_4$, 1 mM $Na_3VO_4$, 5 μg/ml aprotinin, 5 μg/ml leupeptin, and 5 μg/ml pepstatin). The protein concentrations in the cell lysates were determined using the BCA Protein Assay (Thermo Fisher Scientific). Equal amounts of cell lysates were incubated with 1 μg of goat polyclonal antibodies against human Tie1 or Tie2. The immunocomplexes captured by protein G-Sepharose (GE Healthcare) or total cell lysates were separated by SDS-PAGE and transferred to polyvinylidene difluoride membranes. The membranes were probed with rabbit anti-human Tie1, goat anti-human Tie2, mouse anti-phosphotyrosine, rabbit anti-phospho-Akt (S473), anti-Akt, anti-cleaved caspase-3, anti-caspase-3, or mouse anti-α-tubulin antibodies. After washing with TBS containing 0.1% Tween-20, the membranes were probed with HRP-conjugated secondary antibodies and visualized with an ECL or ECL Prime Western Blotting Detection System (GE Healthcare). Chemiluminescent signals were detected with an Amersham Imager 600 (GE Healthcare). For immunoprecipitation of 293-F cell lysates, expression plasmids for full-length Tie1 or Tie2 were transfected using 293fectin (Thermo Fisher Scientific) and grown in serum-free FreeStyle 293 expression medium for 72 h. After the cell number was counted, the cells were resuspended at 2 × $10^6$ cells/ml. Next, 500-μl aliquots of cells were treated with 1 μg/ml Polydom or 500 ng/ml Ang1 and incubated for 15 min at 37°C in a humidified atmosphere containing 8% $CO_2$. The cells were collected by centrifugation and homogenized as described above. Equal amounts of cell lysates were incubated with 1 μg of goat polyclonal antibodies against human Tie1 or Tie2. The immunocomplexes were separated by SDS-PAGE and analyzed by Western blotting as described above.

## Mice

Polydom-deficient mice were generated, bred, and genotyped as described previously (Morooka et al., 2017). All mouse experiments were approved by the Animal Experiment Committee of the Institute for Protein Research, Osaka University (Approval number: R03-01-0) and performed in accordance with institutional guidelines.

## Immunostaining

Whole-mount staining of embryonic mouse skin was performed as described previously (Morooka et al., 2017). Briefly, embryonic mouse back skin was dissected from the underlying musculature and fixed with 4% paraformaldehyde in PBS overnight at 4°C. The fixed tissues were washed three times in PBS containing 0.2% Triton X-100 (PBS-T) for 30 min at 4°C, blocked in PBS-T containing 1% BSA for 1 h at 4°C, and incubated with primary antibodies overnight at 4°C. After six washes in PBS-T for 30 min at 4°C, the bound antibodies were visualized with Alexa Fluor–conjugated secondary antibodies overnight at 4°C. After another four washes in PBS-T for 30 min at 4°C, the tissues were counterstained with PBS-T containing Hoechst 33342 for 30 min at 4°C. After one more wash in PBS-T for 30 min at 4°C, the tissues were flat-mounted on glass slides with FluorSave Reagent (Millipore). For immunostaining of human LECs, cells were seeded on Cell Desk LF1 (Sumitomo Bakelite) in 24-well plates. After overnight serum starvation, the cells were treated with EBM-MV2 containing 0.5% FBS or 1 μg/ml Polydom for 30 min at 37°C in a humidified atmosphere containing 5% $CO_2$. After washing with PBS, the cells were fixed with 3.7% formaldehyde in PBS, permeabilized with PBS containing 0.1% Triton X-100, and blocked with PBS containing 1% BSA. The fixed cells were labeled with anti-human VE-cadherin and rabbit anti-FOXO1 antibodies. The bound antibodies were visualized with Alexa Fluor–conjugated secondary antibodies. Finally, the cells were counterstained with Hoechst33342 and mounted with FluorSave Reagent (Millipore). The stained cells and tissues were imaged using a Fluoview FV1200 confocal microscope (Olympus) with UPlanSApo 20×/0.75 or oil-immersion UPlanSApo 60×/1.30 objective lenses (Olympus). The images were acquired using FV10-ASW software (Olympus) and processed with ImageJ software (http://rsb.info.nih.gov/ij/).

## Flow cytometry

The integrin α9β1 expression levels on LECs were verified by flow cytometry. Suspended LECs were incubated with anti-integrin α9β1 mAb Y9A2 for 1 h at 4°C. Following washing, cells were incubated with Alexa Fluor 488–conjugated goat anti-mouse IgG (H+L) antibody for 30 min at 4°C. Cells were then analyzed on a BD FACSCelesta flow cytometer (BD Biosciences) and BD FACSDiva software (BD Biosciences). Mouse IgG was used as a control. The collected data were analyzed using FlowJo software (FlowJo).

## Statistical analysis

All quantitative data are presented as means ± SD. The statistical significance of differences between paired samples was determined by a two-tailed Student's $t$ test using Microsoft Excel 365. Data were considered statistically significant at $P < 0.05$.

## Online supplemental material

Fig. S1 shows the dose-dependency of Polydom-induced migration of LECs in Transwell assays. Fig. S2 shows that Polydom functions as a chemotactic factor in LEC transmigration. Fig. S3 shows that Polydom does not induce Tie1 phosphorylation. Fig. S4 shows the nuclear internalization of Foxo1 in Polydom-deficient mice. Fig. S5 shows the effects of integrin α9 knockdown or Polydom mutant lacking integrin α9β1 binding activity on LEC migration in Transwell assays.

## Data availability

Original Western blots in Fig. 3, C and F, Fig. 4 B, and Fig. S3, A and B, are available in the supplementary source data files. All other data are available in the published article and its online supplemental material or from the corresponding author (sekiguch@protein.osaka-u.ac.jp) upon reasonable request.

## Acknowledgments

We thank Dr. Mitsuharu Hattori (Nagoya City University, Nagoya, Japan) for providing an expression vector for the Fc region of human immunoglobulin G1 (hIgG1-Fc) and Dr. Yukimasa Taniguchi (Osaka University, Osaka, Japan) for helpful discussions. We also thank Alison Sherwin, PhD, from Edanz (https://jp.edanz.com/ac) for editing a draft of this manuscript.

This work was supported by the Research Fund for Division of Matrixome Research and Application, Institute for Protein Research, Osaka University. This work was also partly supported by Japan Society for the Promotion of Science KAKENHI grants to K. Sekiguchi (25291026) and R. Sato-Nishiuchi (17J40191) and Ministry of Education, Culture, Sports, Science and Technology KAKENHI for Transformative Research Area (A) (23721401). R. Sato-Nishiuchi was the recipient of a Japan Society for the Promotion of Science Restart Postdoctoral Fellowship (2017–2019). Open Access funding provided by Osaka University.

Author contributions: R. Sato-Nishiuchi designed and performed all biochemical analyses (solid-phase binding assays and immunoprecipitation/Western blotting assays) and transmigration/cell adhesion assays, analyzed the data, and wrote the manuscript. M. Doiguchi and N. Morooka analyzed the Polydom-knockout mice and performed the Foxo1 immunostaining. K. Sekiguchi conceptualized and managed the project, analyzed the data, and wrote the manuscript.

Disclosures: K. Sekiguchi reported personal fees from Matrixome, Inc., "other" from Nippi, Inc., grants from Mandom, Inc., and grants from Kao Corporation outside the submitted work. No other disclosures were reported.

Submitted: 10 August 2022

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

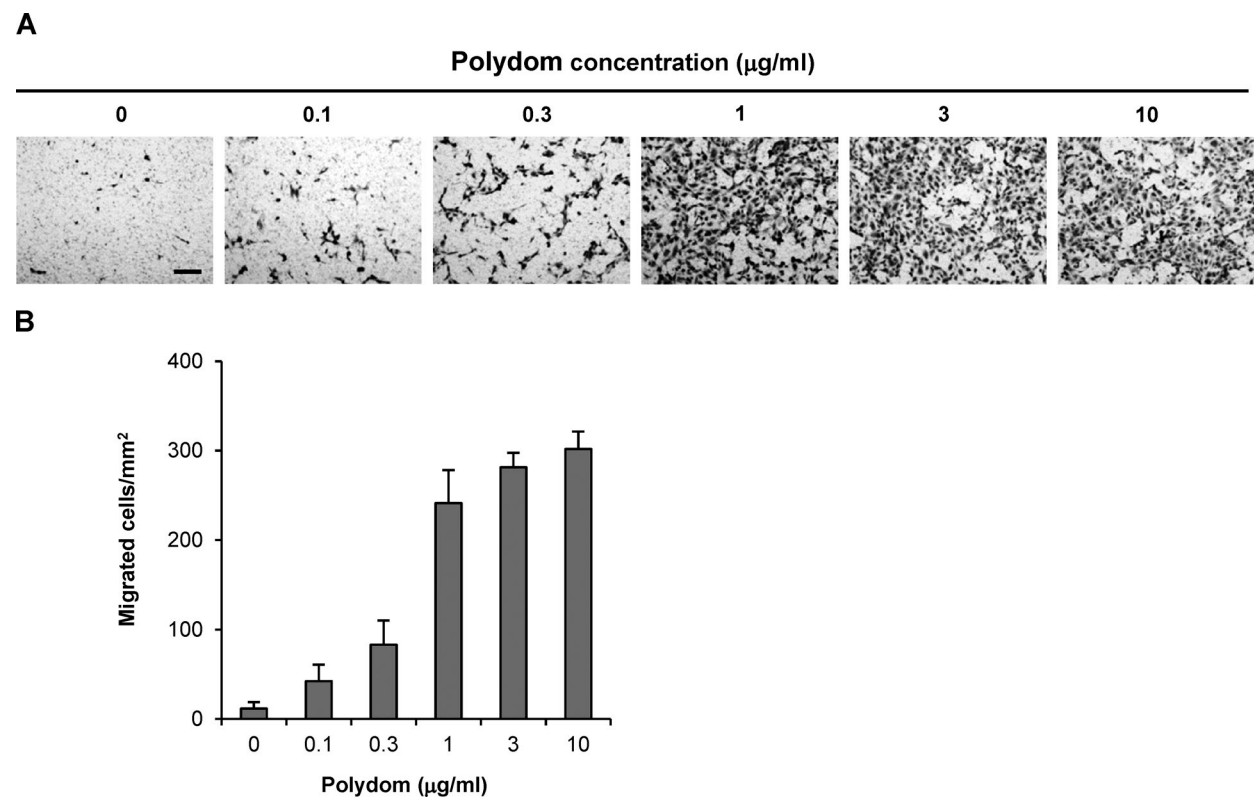

Figure S1. **Polydom promotes LEC migration in a dose-dependent manner. (A)** Photographs of LECs that migrated to the lower side of Transwell membranes when increasing concentrations of Polydom were added to the medium in the lower chamber. Bar, 200 µm. **(B)** Cells that migrated to the lower side of the membranes were counted under a microscope. The data represent means ± SD of three independent experiments each assayed in triplicate.

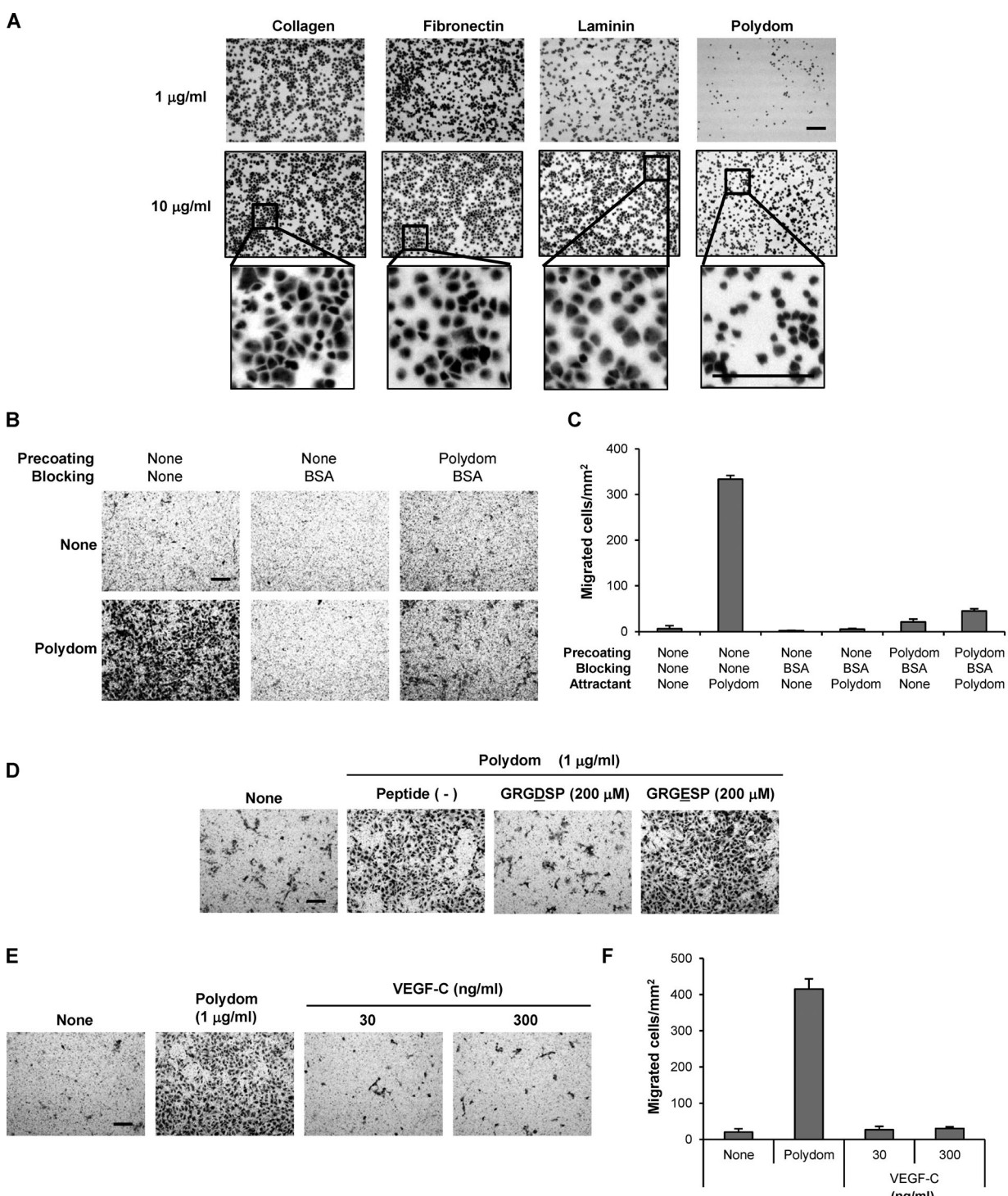

Figure S2. **Polydom functions as a chemotactic, but not a haptotactic, factor in LEC transmigration. (A)** LECs were seeded on 96-well microtiter plates coated with collagen, fibronectin, laminin, or Polydom at 1 or 10 µg/ml and allowed to adhere to the plates for 30 min at 37°C. After washing to remove unbound cells, the attached cells were fixed and stained with toluidine blue. Magnified views of the boxed areas in the middle row are shown in the bottom row. Bars, 200 µm. **(B)** LEC transmigration assays using Transwell membranes with (right panels) or without (left and middle panels) precoating with Polydom (2 µg/ml), followed by blocking with 1% BSA except for the left panels. The upper and lower panels show the results of LEC transmigration in the absence (upper) or presence (lower) of Polydom (1 µg/ml) added to the lower chamber medium. Bar, 200 µm. **(C)** Cells that migrated to the lower side of the membranes were counted under a microscope in the absence (None) or presence of Polydom added to the lower chamber medium. The data represent means ± SD of three independent experiments each assayed in triplicate. **(D)** Cells that migrated to the lower side of the membranes were counted under a microscope in the absence (None) or presence of Polydom (1 µg/ml) with GRGDSP peptide (200 µM) or GRGESP peptide (200 µM). Bar, 200 µm. **(E)** Photographs of LECs that migrated to the lower side of Transwell membranes in the presence of Polydom (1 µg/ml) or VEGF-C (30 or 300 ng/ml) in the lower chamber medium. Bar, 200 µm. **(F)** Cells that migrated to the lower side of the membranes were counted under a microscope in the absence (None) or presence of Polydom or VEGF-C in the lower chamber medium. The data represent means ± SD of three independent experiments each assayed in triplicate.

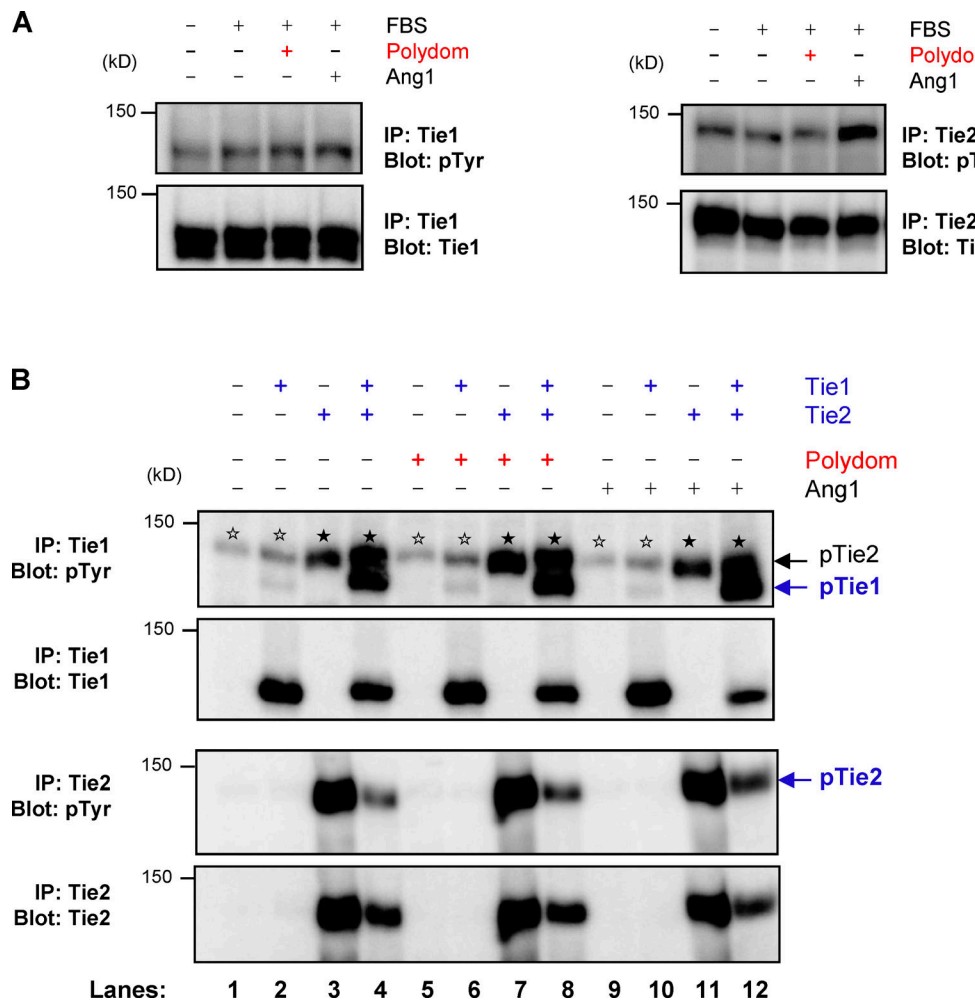

Figure S3.  **Polydom does not induce Tie1 phosphorylation. (A)** Serum-starved LECs were treated with EBM-MV2 medium containing 0.5% FBS, 1 µg/ml Polydom, or 500 ng/ml Ang1 at 37°C for 15 min. Immunoprecipitates (IP) of Tie1 (left) and Tie2 (right) from cell lysates were immunoblotted under reducing conditions for phosphotyrosine residues (upper panels) followed by reimmunoblotting for total Tie1 or Tie2 (lower panels). **(B)** 293-F cells were transfected with the indicated expression plasmids for Tie1 and Tie2 and treated with 1 µg/ml Polydom or 500 ng/ml Ang1 at 37°C for 15 min. Immunoprecipitates of Tie1 and Tie2 from cell lysates were immunoblotted under reducing conditions for phosphotyrosine residues (upper panels), followed by reimmunoblotting for total Tie1 or Tie2 (lower panels). Co-transfection of Tie2 with Tie1 increased Tie1 phosphorylation irrespective of the presence or absence of Polydom (lanes 4, 8, and 12). No Tie1 phosphorylation was induced by Polydom without Tie2 co-transfection (lanes 2, 6, and 10). Signals for phosphorylated Tie2 (pTie2; closed stars) were detected in Tie1 immunoprecipitates from Tie2-transfected cells (uppermost panel; lanes 3, 4, 7, 8, 11, and 12) because the anti-Tie1 polyclonal antibody used for the immunoprecipitation crossreacts with Tie2. Weak signals (open stars) were detected at (or slightly above) the position of pTie2 in the Tie1 immunoprecipitates from cells that were either untransfected or only transfected with Tie1 (lanes 1, 2, 5, 6, 9, and 10). Because Tie2 was not transfected in these cells, the weak signals (open stars) at the pTie2 position should be derived from tyrosine-phosphorylated proteins that were endogenously expressed in 293-F cells and nonspecifically precipitated with the anti-Tie1 antibody used. Such bands were not detected in untransfected or Tie1-transfected cells after immunoprecipitation with an anti-Tie2 antibody (lower panels; lanes 1, 2, 5, 6, 9, and 10). Source data are available for this figure: SourceData FS3.

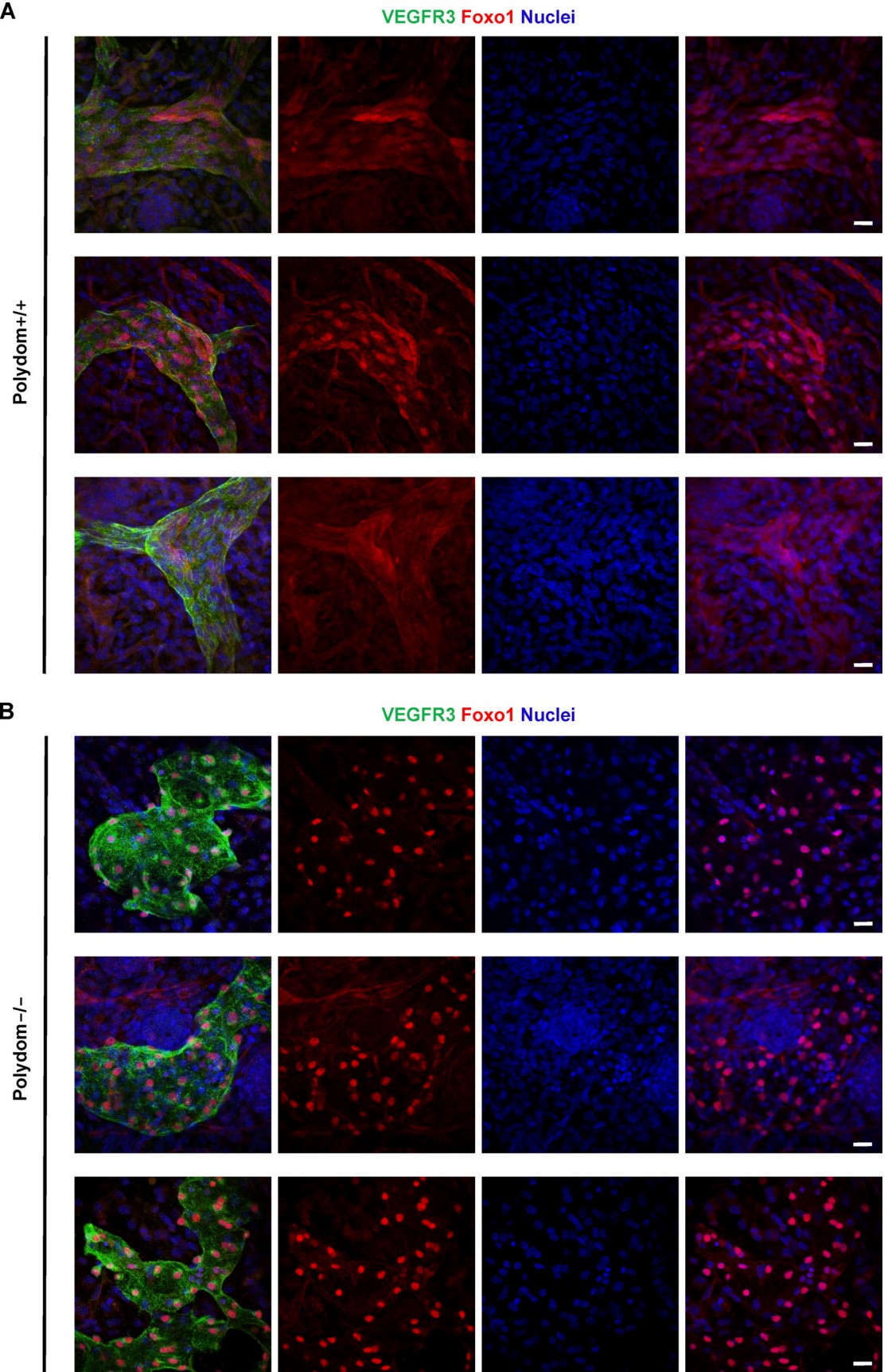

Figure S4.  **Nuclear exclusion of Foxo1 is hampered in Polydom-deficient mice. (A and B)** Whole-mount immunofluorescence staining of E17.5 wild-type (A) and Polydom-deficient mice (B) for VEGFR3 (green) and Foxo1 (red). Nuclei were counter-stained with Hoechst 33342 (blue). Bars, 20 µm.

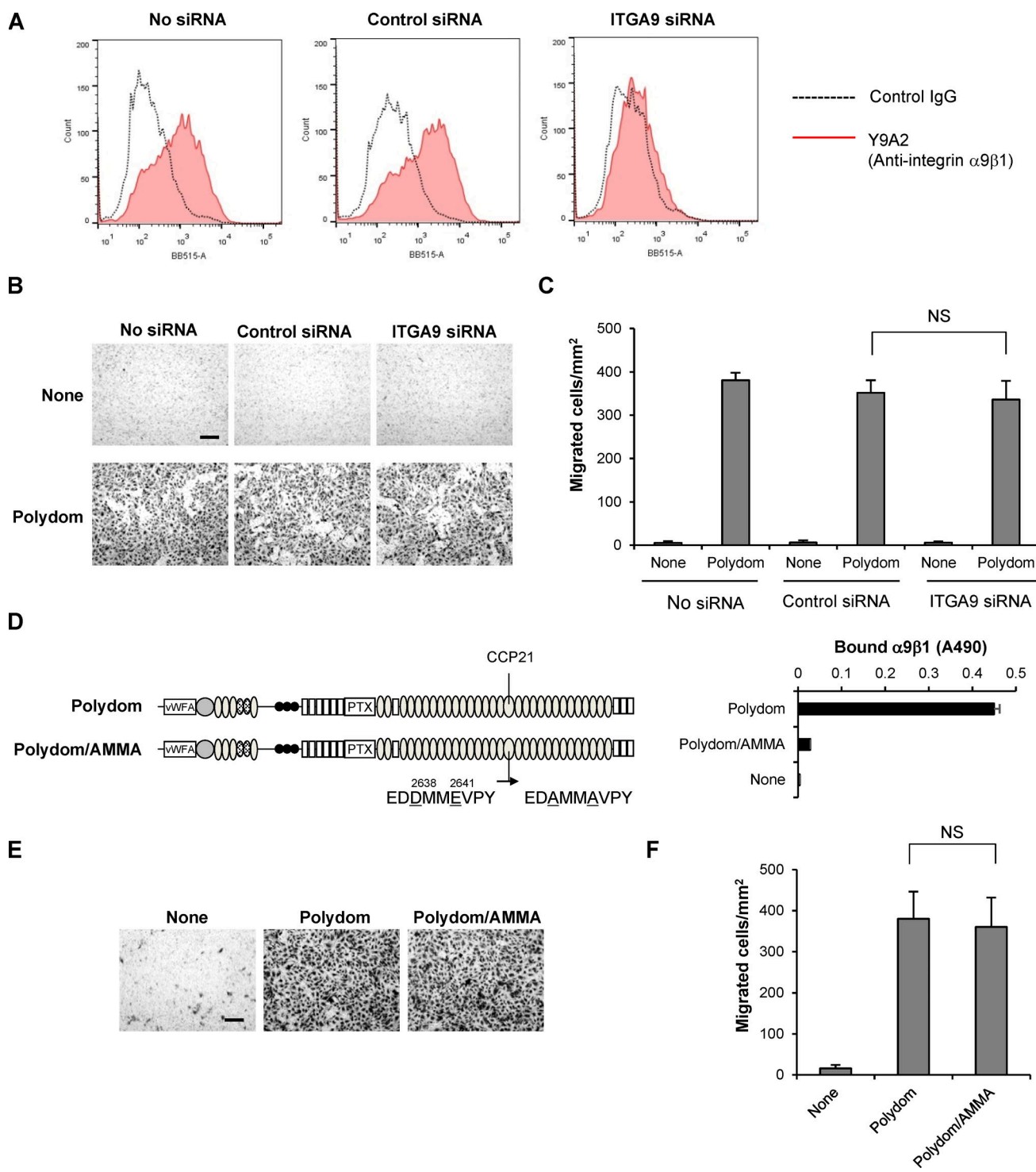

Figure S5. **Integrin α9 is not involved in Polydom-induced LEC migration. (A)** LECs were labeled with an anti-integrin α9β1 mAb (Y9A2; red line) or control mouse IgG (dashed line) and then subjected to flow cytometric analysis. The expression level of integrin α9β1 was significantly reduced in LECs transfected with ITGA9 siRNA. **(B)** Photographs of control and ITGA9 siRNA-treated LECs that migrated to the lower side of Transwell membranes. Bar, 200 μm. **(C)** LECs that migrated to the lower side of the membranes were counted under a microscope. The data represent means ± SD of three independent experiments each assayed in triplicate. NS, not significant (*n* = 3). **(D)** Full-length Polydom and its Asp2638Ala/Glu2641Ala double-mutant (Polydom/AMMA) were assessed for their binding activities toward integrin α9β1. The data represent means ± SD of triplicate determinations. vWFA, von Willebrand factor type A; PTX, pentraxin. **(E)** Photographs of LECs that migrated to the lower side of Transwell membranes in the presence of Polydom or Polydom/AMMA (1 μg/ml) in the lower chamber medium. Bar, 200 μm. **(F)** Cells that migrated to the lower side of the membranes were counted under a microscope. The data represent means ± SD of three independent experiments each assayed in triplicate. NS, not significant (*n* = 3).

