## [Peer Review File · The Journal of Cell Biology]

Polydom/SVEP1 binds to Tie1 and promotes migration of lymphatic endothelial cells

Ryoko Sato-Nishiuchi, Masamichi Doiguchi, Nanami Morooka, and Kiyotoshi Sekiguchi

Corresponding Author(s): Kiyotoshi Sekiguchi, Osaka University

Review Timeline:

Submission Date:	2022-08-10
Editorial Decision:	2022-09-09
Revision Received:	2023-04-13
Editorial Decision:	2023-05-08
Revision Received:	2023-05-11
Editorial Decision:	2023-05-22
Revision Received:	2023-05-26

Monitoring Editor: Tatiana Petrova

Scientific Editor: Tim Fessenden

Transaction Report:

DOI: <https://doi.org/10.1083/jcb.202208047>

September 9, 2022

Re: JCB manuscript #202208047

Prof. Kiyotoshi Sekiguchi
Osaka University
Institute for Protein Research
3-2 Yamadaoka
Suita, Osaka 565-0871
Japan

Dear Prof. Sekiguchi,

Thank you for submitting your manuscript entitled "Polydom/SVEP1 binds to Tie1 and promotes migration of lymphatic endothelial cells". The manuscript has been evaluated by expert reviewers, whose reports are appended below. Unfortunately, after an assessment of the reviewer feedback, our editorial decision is against publication in JCB at this time.

You will see that, while all reviewers agreed that the identification of the ligand for Tie1 is an important advance, all also raised concerns over important details on the consequences of Polydom-Tie1 binding that were missing from this work. In particular, all reviewers stressed that evidence for both intracellular signaling events and lymphatic vessel development *in vivo* were needed to confirm this finding. In addition, Reviewer 3 sought clarification on the additional receptors through which Polydom has been proposed to act. With the exception of Point 2 by Reviewer 3, all comments by all reviewers would need to be addressed.

We feel that the requests made by the reviewers are more substantial than can be addressed in a typical revision period. If you wish to expedite publication of the current data, it may be best to pursue publication at another journal. However, given interest in the topic and the JCB's interest in publishing this work, we would be open to resubmission to JCB of a significantly revised manuscript that fully addresses the reviewers' concerns and is subject to further peer-review. Should you wish to pursue publication with a revised manuscript, please provide a plan for revision. Please note that we may discuss the revision plan with at least one reviewer. If and when you would like to resubmit this work to JCB, please contact the journal office to discuss an appeal of this decision or you may submit an appeal directly through our manuscript submission system.

Regardless of how you choose to proceed, we hope that the comments below will prove constructive as your work progresses. We would be happy to discuss the reviewer comments further once you've had a chance to consider the points raised in this letter. You can contact the journal office with any questions, cellbio@rockefeller.edu or call (212) 327-8588.

Thank you for thinking of JCB as an appropriate place to publish your work.

Sincerely,

Tatiana Petrova
Monitoring Editor
Journal of Cell Biology

Tim Fessenden
Scientific Editor
Journal of Cell Biology

Reviewer #1 (Comments to the Authors (Required)):

This study identifies the extracellular matrix protein Polydom/SVEP1 as a ligand for the endothelial Tie1 receptor. The authors perform biochemical experiments using recombinant proteins and solid-phase binding assays to demonstrate specific interaction of Polydom with Tie1, but not the related Tie2. They further identify the specific amino acids in Polydom that are responsible for Tie1 binding. The biological significance of the interaction is explored in transwell assay where the authors show that Polydom promotes migration of lymphatic endothelial cells (LECs). This response was reduced upon silencing of Tie1, but not Tie2, or if a Polydom mutant that is deficient in Tie1 binding was used.

Tie1 was cloned more than 20 years ago, and its critical functions in the vasculature have been demonstrated, and are still

under active investigation, yet there are no known ligands. The finding presented here is therefore highly significant. The biochemical characterisation of Polydom binding to Tie1 in a solid-phase binding assay is convincing. However, the binding and its consequence on the activity of Tie1 and its downstream pathways are not shown in a cellular context. In addition, the functional evidence is limited, and only based on the transwell migration assay, which in the context of endothelial cells is rather artificial.

Specific questions:

1. Does Polydom binding induce Tie1 phosphorylation, and Tie2 cross-phosphorylation, and downstream signaling?
2. Tie1 is also expressed in blood ECs and regulates blood vessel formation. Did the authors test if Polydom/SVEP1 can also promote migration and Tie1 signaling in these cells? Can they discuss why the phenotype of Polydom deficient mice appears to be restricted to the lymphatic vasculature?
3. Functional evidence for the effect of Polydom-Tie1 interaction on LECs is limited, and based on a rather artificial transwell in vitro migration assay. Scratch wounding assay would be more appropriate.
4. Along the same lines, it might be more relevant to assess if matrix/surface-bound Polydom promotes LEC migration, and how does this compare to e.g., fibronectin? I find it surprising that fibronectin does not promote LEC migration in Fig. 3A and B? Is this because it is not surface-bound?
5. The conclusions would be strongly enhanced by in vivo data, for example virus-mediated overexpression of WT and mutant Polydom in mouse skin, or transgenic expression in zebrafish.

Reviewer #2 (Comments to the Authors (Required)):

The authors present data of the binding of Polydom to Tie1, specify the domain mediating the binding and substitute residues in Polydom abrogating the binding. They conclude using a Transwell assay that Polydom, via Tie1, induces lymphatic endothelial cell migration. The findings are potentially interesting, but experimentation is limited and as such not sufficient to conclude that Polydom is a Tie1 ligand. The authors motivate their studies by similar lymphovascular phenotypes of Polydom and Tie1 deficient mouse embryos. However, since the manuscript contains no data from any in vivo models, this speculation should be limited to discussion.

Major comments:

1. The results from solid phase binding assays provide strong support that Tie1 can bind to Polydom in vitro, but no experiments in vivo are presented to support the conclusion that Polydom is a ligand for Tie1 in the lymphatic vasculature.
2. Therefore, in several places the statements of the physiological role of Polydom as a Tie1 ligand during lymphatic development are not supported by data. E.g. last sentence in the abstract: "The present findings indicate that Polydom is a physiological ligand for Tie1 and participates in lymphatic vessel development by promoting lymphatic endothelial cell migration through its interaction with Tie1"
3. To support the function of Polydom as a ligand for Tie1, the consequences of Polydom binding to Tie1 and subsequent downstream signalling and receptor tyrosine kinase activation should be investigated.
4. In alanine scanning mutagenesis, authors do not provide data on the stability of the mutant Polydom or its domains, which might affect the overall structural/biophysical properties of the protein and thus, indirectly binding to Tie1. Thus, it remains open, if the identified amino acids are required at the protein-protein interaction interface or if they have a structural role in maintaining the overall stability of the 3D structure.
5. If Region 2 mediates the binding of Polydom to Tie1, a Region2-derived peptide should inhibit the binding.
6. The mechanism behind Polydom-induced lymphatic endothelial cell migration in the Transwell assay requires further investigation. The authors should use a positive control, potentially VEGF-C to assess the magnitude of Polydom-induced migration. Does Polydom increase the adhesion of lymphatic ECs, evident by comparing the cell numbers in the bottom of the filter and bottom of the well? The results concerning cell migration would be much strengthened using live imaging.
7. Additional controls would strengthen the Transwell cell migration assay. It should be investigated, if silencing of Tie1 impairs lymphatic EC migration in general or in response to also other chemotactic factors than Polydom? It should also be validated that Tie1 siRNApool does not affect lymphatic EC survival, which could affect cell migration.

8. The authors consider that "given the essential role of integrins in cell migration, it is tempting to speculate that cooperative interactions between Polydom, Tie1, Ang2, and integrins facilitate the lymphatic endothelial cell migration required for remodeling of lymphatic vessels." However, this is not supported by the results since silencing of Itga9 did not affect lymphatic EC migration in response to Polydom in Supplemental Fig. S2.

Other comments:

9. Throughout the text, it is emphasised that the phenotypes of Tie1 deficient embryos are similar to those in Polycomb deficient mice especially considering lymphatic vascular remodelling and the formation of valves (Qu et al., 2010; Shen et al., 2014). However, the authors do not consider the earlier abnormal formation of jugular lymph sacs in Tie1 mutants (D'Amico et al ATVB 2010).

10. Rabbit anti-human Tie1 (C-18) has been discontinued and details of it cannot be found, e.g. potential cross reactivity with Tie2, which should be mentioned.

11. Based on the materials and methods, it is unclear in which media the cells were cultured for the Transwell assays and what is meant by starvation in this experiment.

12. It would be helpful to compare the treatments if molar concentrations are provided in Fig. 3A.

Reviewer #3 (Comments to the Authors (Required)):

In this study the authors extend prior work on the role of the matrix protein Polydom during lymphatic vessel growth and development. Polydom is required for lymphatic vessel growth, maturation and valve formation, but the molecular mechanism by which it regulates lymphatic development remains unclear. Prior work by these authors identified Polydom as a high affinity ligand for integrin $\alpha 9\beta 1$, an adhesive receptor known to participate in lymphatic valve formation and vessel maturation. Subsequent studies in fish and mice established a conserved role for Polydom in lymphatic development. The authors also previously reported that Polydom binds Ang1 and 2, suggesting that its role in lymphatic growth may be related to Angiopoietin activity as well as $\alpha 9\beta 1$ adhesion. In the present study the authors identify binding of Polydom to Tie1 and use in vitro studies to demonstrate that a Polydom mutant unable to bind Tie1 is unable to stimulate LEC migration like wild-type Polydom. They conclude from these studies that Polydom is a Tie1 ligand and that its effects on LEC migration - and by extension lymphatic development? - are mediated by Tie1. Unfortunately, the data presented in the paper do not strongly support this conclusion and fail to explain (i) how Polydom-Tie1 binding controls migration and LEC function, and (ii) how Polydom binding to $\alpha 9\beta 1$ and Ang ligands contributes to its function if Tie1 binding is required. The study is very preliminary and requires more rigorous biochemical and functional studies to fully test the hypothesis that Polydom regulates lymphatic maturation and growth through Tie1 ligand activity.

Major points:

1. Does Polydom stimulate Tie1 signaling? The authors present binding data that support Polydom-Tie1 interaction and suggest that these findings identify Polydom as a novel Tie1 ligand that is required for lymphatic growth and maturation. Presumably such an effect would be mediated by Tie1 signal transduction, esp since they provide data excluding Tie2, but the authors fail to state this overtly and do not measure any signaling effects of Polydom. Studies testing Polydom stimulation of Tie1 phosphorylation and downstream signaling effectors such as ERK1/2 and PI3K/AKT/mTOR is essential to address their hypothesis.

2. Role of $\alpha 9\beta 1$ binding. The authors use the Polydom E2567A mutant as evidence that Polydom's effect on LEC migration is Tie1-mediated. Does the E2567A mutant impact integrin $\alpha 9\beta 1$ binding? If yes, their conclusion would appear over-simplified and not complete. If no, how do the authors reconcile their findings with prior work demonstrating strong $\alpha 9\beta 1$ binding and the fact that loss of both Polydom and $\alpha 9\beta 1$ result in lymphatic valve defects in vivo? The argument that $\alpha 9\beta 1$ is not the relevant receptor for Polydom because the phenotype of the Polydom mutant and the Itga9 mutant are not precisely identical is not a convincing one because there may be redundancy for integrin/adhesion receptor binding of Polydom.

3. Role of Angiopoietin binding. The authors have also previously reported that Polydom binds Ang1 and Ang2, and argue in the discussion that this is not relevant to the present findings because Tie2 knockdown did not block Transwell migration of LECs stimulated by Polydom. This argument further suggests that they are proposing that Polydom stimulates Tie1 signaling, so direct signaling assays are essential. Also, is it possible that Tie1 binding is mediated by Polydom-bound Angiopoietin ligands rather than a direct Tie1-Polydom interaction? The fact that the authors have now identified 4 binding partners for Polydom makes this biology less rather than more transparent.

4. In vivo correlation. The authors provide no in vivo data to support the proposed Tie1 mechanism. The functional studies are

limited to Transwell migration assays that may or may not be relevant to the major in vivo phenotypes observed with loss of Polydom in vivo. Supportive in vivo data would help clarify the mechanism.

Minor points:

Throughout the manuscript the authors fail to clearly state how they believe Polydom-Tie1 interaction regulates LEC migration at the molecular level. This needs to be stated in a clear and forthright manner, e.g. accompanied by a diagram that fully explains the authors' proposed molecular mechanism.

Reviewer #1

This study identifies the extracellular matrix protein Polydom/SVEP1 as a ligand for the endothelial Tie1 receptor. The authors perform biochemical experiments using recombinant proteins and solid-phase binding assays to demonstrate specific interaction of Polydom with Tie1, but not the related Tie2. They further identify the specific amino acids in Polydom that are responsible for Tie1 binding. The biological significance of the interaction is explored in transwell assay where the authors show that Polydom promotes migration of lymphatic endothelial cells (LECs). This response was reduced upon silencing of Tie1, but not Tie2, or if a Polydom mutant that is deficient in Tie1 binding was used.

Tie1 was cloned more than 20 years ago, and its critical functions in the vasculature have been demonstrated, and are still under active investigation, yet there are no known ligands. The finding presented here is therefore highly significant. The biochemical characterisation of Polydom binding to Tie1 in a solid-phase binding assay is convincing. However, the binding and its consequence on the activity of Tie1 and its downstream pathways are not shown in a cellular context. In addition, the functional evidence is limited, and only based on the transwell migration assay, which in the context of endothelial cells is rather artificial.

We thank the reviewer for the encouraging and constructive comments on our manuscript. We agree with the reviewer that our manuscript relies on the Transwell migration assays to demonstrate the physiological relevance of Polydom binding to Tie1. To address this critique, we have performed additional experiments to explore the signaling events elicited by Polydom binding to Tie1, as detailed in the responses below.

Specific questions:

1. Does Polydom binding induce Tie1 phosphorylation, and Tie2 cross-phosphorylation, and downstream signaling?

We are fully aware of the importance of examining whether Polydom binding to Tie1 induces Tie1 phosphorylation and Tie2 cross-phosphorylation. Thus, we repeatedly performed Tie1/Tie2 phosphorylation assays by immunoprecipitation of Tie1/Tie2 from lysates of Polydom-treated LECs, followed by immunodetection of tyrosine-phosphorylated Tie1/Tie2. However, we did not obtain any convincing evidence for Tie1 or Tie2 phosphorylation in Polydom-treated LECs (**new Supplementary Fig. S3 A**); we did observe a very small increase in phosphorylated Tie1 signals once or twice, but this small signal increase was not reproduced in more than 10 other independent experiments. We therefore concluded that Tie1 ligation by Polydom does not increase the phosphorylation of Tie1 or Tie2 in LECs. This conclusion was supported by another experiment in which 293-F cells were transfected with full-length Tie1 and/or Tie2, followed by Polydom treatment. Again, no significant Tie1 phosphorylation was induced by Polydom in the Tie1-transfected cells (**new Supplementary Fig. S3 B**). Notably, Tie1 phosphorylation was induced in cells co-transfected with Tie1 and Tie2 irrespective of the presence or absence of Polydom, consistent with previous observations that Tie1 was only phosphorylated in 293 cells when Tie1 and Tie2 were co-expressed (Saharinen et al., J Cell Biol, 2005;169:239–243; Yuan et al., FASEB J, 2007;21:3171–3183; Korhonen et al., J Clin Invest, 2016;126:3495–3510).

It is well known that the PI3K/Akt signaling pathway becomes activated downstream of the Ang-Tie axis. We therefore examined whether the PI3K/Akt pathway was activated in LECs following Tie1 ligation by Polydom. We found that Polydom induced Akt phosphorylation in LECs in a Tie1-dependent manner (**new Fig. 4 B**), indicating involvement of the PI3K/Akt pathway in the Polydom-

induced LEC migration. In support of this notion, Polydom-induced transmigration was inhibited by two PI3K inhibitors, LY294002 and Wortmannin, but not by an ERK inhibitor (**new Fig. 4 A**). In addition, we observed that nuclear exclusion of FOXO1, a signaling event elicited downstream of Akt activation, was induced in LECs after Polydom treatment (**new Fig. 4, C and D**). These new data have been added to the revised manuscript (page 7, lines 182–196; page 8, lines 221–232).

2. Tie1 is also expressed in blood ECs and regulates blood vessel formation. Did the authors test if Polydom/SVEP1 can also promote migration and Tie1 signaling in these cells? Can they discuss why the phenotype of Polydom deficient mice appears to be restricted to the lymphatic vasculature?

We thank the reviewer for raising this point. We performed the Transwell migration assay with HUVECs and found that Polydom promoted the transmigration of HUVECs, similar to the case for LECs (please see data below).

Polydom promotes HUVEC (human umbilical vascular endothelial cell) migration

Regarding the phenotype of Polydom-deficient mice, particularly the phenotype for blood vessel formation, we previously observed defects in venous development in the skin of E15.5 Polydom-deficient mice (Online Figure II in Morooka et al., *Circ Res*, 2017), wherein alignment of larger-diameter vessels was not detected. Such defects in venous formation were also observed in the intercostal region (please see data below), but not in the cardinal vein or mesenteries. Similar defects in venous formation in the skin were recently reported for Tie1-deficient mice (Cao et al., *bioRxiv* 2022.08.05.502976; <https://doi.org/10.1101/2022.08.05.502976>). The phenotypes for venous formation in Polydom-knockout and Tie1-knockout mice are discussed, with reference to the phenotypes of svep1/polydom-deficient and tie1-deficient zebrafish reported by Hußmann et al. (*bioRxiv* 2022.09.28.509871; <https://doi.org/10.1101/2022.09.28.509871>), in the revised manuscript (page 10, line 288, through page 11, line 296).

3. Functional evidence for the effect of Polydom-Tie1 interaction on LECs is limited, and based on a rather artificial transwell in vitro migration assay. Scratch wounding assay would be more appropriate.

We thank the reviewer for raising this point. As detailed in our response to the reviewer's comment 4 below, our Transwell migration assay seemed to reflect chemotactic, but not haptotactic, activity of Polydom, because Polydom only induced LEC transmigration when it was added to the lower chamber medium and not when it was coated on the lower surface of the Transwell membrane (**new Supplementary Fig. S2, B and C**). The intrinsic cell-adhesive activity of Polydom toward LECs was rather weak compared with that of fibronectin, collagen, and laminin (**new Supplementary Fig. S2 A**). We tried to conduct a scratch wound assay but failed to detect any cell migration-promoting activity of Polydom.

4. Along the same lines, it might be more relevant to assess if matrix/surface-bound Polydom promotes LEC migration, and how does this compare to e.g., fibronectin? I find it surprising that fibronectin does not promote LEC migration in Fig. 3A and B? Is this because it is not surface-bound?

As described above, Polydom did not promote LEC migration when coated on the lower surface of the Transwell membrane followed by BSA blocking (**new Supplementary Fig. S2, B and C**), even when Polydom was added to the lower chamber medium, arguing against its function as a matrix/surface-bound haptotactic factor. It should be noted that cell migration is driven by integrin-mediated cell-matrix interactions. Thus, LECs did not transmigrate when the lower surface of the Transwell membrane was blocked by BSA without any pre-coating, even when Polydom was added to the lower chamber medium. Because the medium contains 0.5% FBS, it is conceivable that cell-adhesive protein(s) present in FBS were adsorbed onto the lower surface of the Transwell

membrane, thereby providing adhesive substrates for LEC transmigration. In support of this notion, an RGD-containing peptide inhibited the Polydom-induced transmigration (**new Supplementary Fig. S2 D**), suggesting that fibronectin, vitronectin, or other RGD-containing proteins in FBS provide cell-adhesive substrates for transmigrating LECs. These new data have been added to the revised manuscript (page 5, line 143 through page 6, line 152).

5. The conclusions would be strongly enhanced by in vivo data, for example virus-mediated overexpression of WT and mutant Polydom in mouse skin, or transgenic expression in zebrafish.

We agree with the reviewer that our conclusions would be strongly enhanced by in vivo data. We attempted to package the full-length Polydom cDNA into a lentiviral vector. However, the yield of the recombinant virus was extremely low because the 10.7-kb length of the full-length Polydom cDNA exceeds the optimal packaging capacity of commercially available viral vectors. In addition, transgenic expression in zebrafish is not available in our laboratory, making it difficult to obtain such data for the revised manuscript. However, given the involvement of the PI3K/Akt signaling pathway downstream of the Polydom-Tie1 interaction and the nuclear exclusion of FOXO1 in Polydom-treated LECs, we sought to determine whether the activation/inactivation status of Foxo1 is altered in Polydom-deficient mice. We found that Foxo1 was exclusively localized within the nucleus in Polydom-deficient mice, while it was detected in both the nucleus and cytoplasm in wild-type mice, with significantly reduced signal intensity in the nucleus of the wild-type mice compared with the Polydom-deficient mice (**new Fig. 4, E and F**). These data provide in vivo evidence for the physiological significance of Polydom binding to Tie1 (page 7, lines 196–203).

Recently, Stefan Schulte-Merker and his group deposited a preprint in bioRxiv entitled "svep1 and tie1 genetically interact and affect aspects of facial lymphatic development in a Vegfc-independent manner" (Hußmann et al., 2022; <https://doi.org/10.1101/2022.09.28.509871>). This paper provides genetic evidence for an interaction of svep1/polydom with tie1 based on similarities in the phenotypes of svep1-knockout and tie1-knockout zebrafish, further supporting a physiological function for the Polydom-Tie1 interaction in lymphangiogenesis (page 10, line 288 through page 11, line 296).

Reviewer #2 (Comments to the Authors (Required)):

The authors present data of the binding of Polydom to Tie1, specify the domain mediating the binding and substitute residues in Polydom abrogating the binding. They conclude using a Transwell assay that Polydom, via Tie1, induces lymphatic endothelial cell migration. The findings are potentially interesting, but experimentation is limited and as such not sufficient to conclude that Polydom is a Tie1 ligand. The authors motivate their studies by similar lymphovascular phenotypes of Polydom and Tie1 deficient mouse embryos. However, since the manuscript contains no data from any in vivo models, this speculation should be limited to discussion.

We thank the reviewer for the useful and constructive comments. We agree with the reviewer that our experimental findings were limited for the assertion that Polydom is a physiological ligand for Tie1 in the absence of any in vivo data in the original manuscript. In the revised manuscript, we have gathered more data to support our conclusion by addressing the signaling events elicited by Polydom in vitro and in vivo, as detailed in the responses below.

Major comments:

1. The results from solid phase binding assays provide strong support that Tie1 can bind to Polydom in vitro, but no experiments in vivo are presented to support the conclusion that Polydom is a ligand for Tie1 in the lymphatic vasculature.

We appreciate this comment on the importance of in vivo data. In the present study, we demonstrated that Polydom specifically bound to Tie1 and promoted lymphatic endothelial cell (LEC) migration in Transwell assays. In a typical reverse genetics approach, the next step would be the production of Polydom-knockout mice and comparison of their phenotype with that of Tie1-knockout mice to explore the physiological relevance of the molecular interaction between Polydom and Tie1 in vivo. In reality, we had already produced Polydom-knockout mice and found that these mice suffer from defects in lymphatic vessel remodeling and die immediately after birth. As expected, this phenotype of Polydom-knockout mice resembles the phenotype of Tie1-knockout mice, which also exhibit defects in lymphatic vessel remodeling. Thus, from a reverse genetics approach, the physiological significance of the Polydom-Tie1 interaction is at least partially supported by in vivo genetics. The phenotype of the Polydom-knockout mice was published in a previous report (Morooka et al., *Circ Res*, 2017;120:1276–1288), and therefore we did not include any overlapping in vivo data derived from these mutant mice in the original manuscript.

Nevertheless, we agree with the reviewer that we did not present any in vivo experiments in our manuscript to support the conclusion that Polydom is a ligand for Tie1 in the lymphatic vasculature. Accordingly, we sought to obtain additional data that would support our conclusion. Given the phenotype of the Polydom-knockout mice, we explored the signaling events elicited downstream of the Polydom-Tie1 interaction. We found that Polydom activated the PI3K/Akt signaling pathway and induced nuclear exclusion of FOXO1 (**new Fig. 4, B–D**), a signaling event induced downstream of the PI3K/Akt pathway in LECs and blood endothelial cells (Kontos et al., *Mol Cell Biol*, 1998;18:4131–4140; Kim et al., *Circ Res*, 2000;86:24–29; Korhonen et al., *J Clin Invest*, 2016;126:3495–3510). We therefore examined the Foxo1 localization by immunostaining the lymphatic vessels in Polydom-knockout mice. We found that Foxo1 was exclusively localized within the nucleus in Polydom knockout mice, while it was detected in both the nucleus and cytoplasm in wild-type mice, with significantly reduced signal intensity in the nucleus of the wild-type mice compared with the Polydom-deficient mice (**new Fig. 4, E and F**). These data provide in vivo evidence for the involvement of Polydom in the signaling events elicited downstream of Tie1 (page 7, lines 191–203).

Recently, Stefan Schulte-Merker and his group deposited a preprint in bioRxiv entitled "svep1 and tie1 genetically interact and affect aspects of facial lymphatic development in a Vegfc-independent manner" (Hußmann et al., 2022; <https://doi.org/10.1101/2022.09.28.509871>). This paper provides genetic evidence for an interaction of svep1/polydom with tie1 based on similarities in the phenotypes of svep1-knockout and tie1-knockout zebrafish, further supporting a physiological function for the Polydom-Tie1 interaction in lymphangiogenesis (page 10, line 288 through page 11, line 296).

2. Therefore, in several places the statements of the physiological role of Polydom as a Tie1 ligand during lymphatic development are not supported by data. E.g. last sentence in the abstract: "The present findings indicate that Polydom is a physiological ligand for Tie1 and participates in lymphatic vessel development by promoting lymphatic endothelial cell migration through its interaction with Tie1"

As detailed in our response to the reviewer's comment 1, we have added new data showing the Polydom-induced activation of the PI3K/Akt signaling pathway and subsequent exclusion of Foxo1 from the nucleus of LECs both in vitro and in vivo. We believe that these new data support our conclusion that Polydom binding to Tie1 is physiologically relevant to lymphatic vessel remodeling, as implied by the similarities in the phenotypes of Polydom-knockout and Tie1-knockout mice.

3. To support the function of Polydom as a ligand for Tie1, the consequences of Polydom binding to Tie1 and subsequent downstream signalling and receptor tyrosine kinase activation should be investigated.

We appreciate this comment. We are fully aware of the importance of examining whether Polydom binding to Tie1 induces Tie1 phosphorylation. Thus, we repeatedly performed Tie1/Tie2 phosphorylation assays by immunoprecipitation of Tie1/Tie2 from lysates of Polydom-treated LECs, followed by immunodetection of tyrosine-phosphorylated Tie1/Tie2. However, we did not obtain any convincing evidence for Tie1 or Tie2 phosphorylation in Polydom-treated LECs (**new Supplementary Fig. S3 A**); we did observe a very small increase in phosphorylated Tie1 signals once or twice, but this small signal increase was not reproduced in more than 10 other independent experiments. We therefore concluded that Tie1 ligation by Polydom does not induce the phosphorylation of either Tie1 or Tie2 in LECs. This conclusion was supported by another experiment in which 293-F cells were transfected with full-length Tie1 and/or Tie2, followed by Polydom treatment. Again, no significant Tie1 phosphorylation was induced by Polydom in the Tie1-transfected cells (**new Supplementary Fig. S3 B**). Notably, Tie1 phosphorylation was induced in cells co-transfected with Tie1 and Tie2 irrespective of the presence or absence of Polydom, consistent with previous observations that Tie1 was only phosphorylated in 293 cells when Tie1 and Tie2 were co-expressed (Saharinen et al., J Cell Biol, 2005;169:239–243. 2005; Yuan et al., FASEB J, 2007;21:3171–3183; Korhonen et al., J Clin Invest, 2016;126:3495–3510). These data have been included in the revised manuscript (page 6, line 172 through page 7, line 181).

4. In alanine scanning mutagenesis, authors do not provide data on the stability of the mutant Polydom or its domains, which might affect the overall structural/biophysical properties of the protein and thus, indirectly binding to Tie1. Thus, it remains open, if the identified amino acids are required at the protein-protein interaction interface or if they have a structural role in maintaining

the overall stability of the 3D structure.

We thank the reviewer for raising this point. Stability of recombinant proteins is an important concern when mutant proteins harboring amino acid mutations and/or deletions are expressed using mammalian expression systems. We have produced hundreds of mutant proteins for not only Polydom but also other extracellular matrix proteins, such as laminins (e.g., Takizawa et al., *Sci Adv*, 2017;3:e1701497; Taniguchi et al., *Matrix Biol*, 2020;87:66–76), to locate the amino acid residues involved in protein-protein interactions. From these experiments, we have learned that the yield of recombinant proteins is a good indication for their proper folding and therefore their stability. Specifically, any mutation that has a deteriorative impact on protein folding significantly reduces the yield of the recombinant protein. The yield of the Polydom mutant that did not bind to Tie1 was comparable to that of intact Polydom, making it unlikely that the introduced mutation interfered with the recombinant protein folding and stability or its secretion. We also constructed a 3D structure model for the CCP20 domain harboring the Tie1-binding site using AlphaFold-2. The critical residues for Tie1 binding, Glu2567 and Gly2568, were found to be exposed on the surface of the CCP20 domain (see predicted 3D structure of CCP20 below). Thus, it seems unlikely that mutations at these residues affect the stability of the associated recombinant proteins or cause non-physiological protein-protein interactions.

5. If Region 2 mediates the binding of Polydom to Tie1, a Region2-derived peptide should inhibit the binding.

We appreciate this comment. It will be interesting to examine whether a Region-2-derived peptide can inhibit Polydom binding to Tie1 and what effect it would have on the IC50. As shown in **Fig. 2 B**, a CCP22 mutant in which Region-2 was replaced with that of CCP20 (CCP22/20-R2) exhibited significantly reduced Tie1-binding activity compared with intact CCP20, suggesting that the Tie1-binding activity is highly conformation-dependent. Although we have not examined the inhibitory potency of a Region-2 peptide, a high concentration of the peptide may be required to interfere with the Polydom-induced LEC migration. We will continue to seek therapeutic peptides or small recombinant proteins that can modulate the Polydom-Tie1 interaction in lymphangiogenesis under pathological conditions.

6. The mechanism behind Polydom-induced lymphatic endothelial cell migration in the Transwell assay requires further investigation. The authors should use a positive control, potentially VEGF-C to assess the magnitude of Polydom-induced migration. Does Polydom increase the adhesion

of lymphatic ECs, evident by comparing the cell numbers in the bottom of the filter and bottom of the well? The results concerning cell migration would be much strengthened using live imaging.

We appreciate this comment. We examined the capability of VEGF-C to promote LEC migration in our Transwell assay. To our surprise, VEGF-C did not promote LEC transmigration even at the concentration of 300 ng/ml (**Fig. S2, E and F**), suggesting that the mechanism for Polydom-induced LEC migration differs from that for VEGF-C-induced sprouting of LEC progenitors in the initial stage of lymphatic vessel development (page 6, lines 152–156).

We also examined the cell-adhesive activity of Polydom toward LECs. Although Polydom was very potent in promoting LEC transmigration, it was much less active than collagen, fibronectin, and laminin in inducing adhesion and subsequent spreading of LECs in a standard cell adhesion assay (**new Fig. S2 A**). Furthermore, when the lower side of the Transwell membrane was precoated with Polydom followed by blocking with BSA, LEC transmigration was not observed even when Polydom was added to the medium in the lower chamber (**new Fig. S2, B and C**). Because the lower chamber medium contains 0.5% FBS, it is conceivable that cell-adhesive protein(s) in FBS were adsorbed on the lower surface of the Transwell membrane, thereby providing adhesive substrates for LEC migration. In support of this notion, an RGD-containing peptide inhibited the Polydom-induced transmigration (**new Supplementary Fig. S2 D**), suggesting that fibronectin, vitronectin, or other RGD-containing proteins present in FBS provide cell-adhesive substrates for transmigrating LECs. These new results have been described in the revised manuscript (page 5, line 143 through page 6, line 152).

We performed preliminary experiments for live imaging of LECs on glass-bottom dishes precoated with 0.5% FBS in the presence or absence of Polydom in the medium. We found that Polydom increased membrane ruffles in LECs, an indication of enhanced cell migration. We will pursue the mechanistic basis for Polydom-induced LEC migration in a future study.

7. Additional controls would strengthen the Transwell cell migration assay. It should be investigated, if silencing of Tie1 impairs lymphatic EC migration in general or in response to also other chemotactic factors than Polydom? It should also be validated that Tie1 siRNApool does not affect lymphatic EC survival, which could affect cell migration.

As described above, VEGF-C did not promote LEC migration in our Transwell assay. We further examined whether Tie1 siRNA transfection had an impact on LEC survival using a cleaved caspase-3 assay. As shown in the **new Fig. 3 F**, no cleaved caspase-3, a marker for apoptotic cells, was detected in siRNA-treated LECs (page 6, lines 161–162).

8. The authors consider that "given the essential role of integrins in cell migration, it is tempting to speculate that cooperative interactions between Polydom, Tie1, Ang2, and integrins facilitate the lymphatic endothelial cell migration required for remodeling of lymphatic vessels." However, this is not supported by the results since silencing of Itga9 did not affect lymphatic EC migration in response to Polydom in Supplemental Fig. S2.

We thank the reviewer for this comment. We apologize for the insufficient discussion on the role of integrins in Polydom-induced LEC migration. As indicated by the reviewer, siRNA-mediated knockdown of Itga9 did not affect LEC migration in response to Polydom (**Supplementary Fig. S5 A–C**). We further confirmed that integrin alpha9beta1 was not involved in Polydom-induced LEC migration by producing a full-length Polydom mutant in which the integrin alpha9beta1 recognition sequence EDDMMEVPY was substituted with EDAMMAVPY. The mutant protein was fully active in promoting LEC transmigration, even though it was inactive in binding to integrin alpha9beta1

(new Fig. S5, D–F), thereby excluding the possibility that integrin alpha9beta1 is involved in Polydom-induce LEC migration (page 9, lines 255–258).

LECs express a variety of other integrins on their surface. We comprehensively analyzed the integrins expressed on LECs by flow cytometry (please see data below). Integrins alpha1, alpha2, alpha3, alpha5, and alpha6, all of which can heterodimerize with integrins beta1, alphavbeta3, and alphavbeta5, were abundantly expressed on LECs. Because Tie1 and Tie2 were reported to associate with integrins alpha5beta1 and alphavbeta3 (Cascone et al., *J Cell Biol*, 2005;170:993–1004; Dalton et al., *PLoS One*, 2016;11:e0163732) and thereby modulate the PI3K/Akt pathway in an integrin-dependent manner (Korhonen et al., *J Clin Invest*, 2016;126:3495–3510), it is tempting to speculate that these integrins are involved in activation of the PI3K/Akt signaling pathway downstream of the Polydom-Tie1 interaction. Essential roles for integrins in the activation of intracellular signaling cascades, including the PI3K/Akt pathway, have been well documented. In support of this notion, Polydom-induced LEC migration was abrogated by an RGD-containing peptide, indicating that RGD-binding integrins are involved in the migration. The above discussion has been added to the revised manuscript (page 9, lines 236–249).

Other comments:

9. Throughout the text, it is emphasised that the phenotypes of Tie1 deficient embryos are similar to those in Polycomb deficient mice especially considering lymphatic vascular remodelling and the formation of valves (Qu et al., 2010; Shen et al., 2014). However, the authors do not consider the earlier abnormal formation of jugular lymph sacs in Tie1 mutants (D'Amico et al ATVB 2010).

We thank the reviewer for raising this point. In our previous paper (Morooka et al., Circ Res, 2017;120:1276–1288), we described that no apparent defects in Prox1-positive primordial thoracic duct formation were observed at E12.5 (Online Figure III), suggesting that the specification of LECs and the formation of the first lymphatic structures are not affected by disruption of *Polydom*. It is interesting to note the report by Shen et al. (Arterioscler Thromb Vasc Biol, 2014;34:1221–1230), which indicated that lymph sac formation occurred without obvious abnormality in Tie1-deficient mice, consistent with our observations for Polydom knockout mice.

10. Rabbit anti-human Tie1 (C-18) has been discontinued and details of it cannot be found, e.g. potential cross reactivity with Tie2, which should be mentioned.

We thank the reviewer for raising this issue. As indicated by the reviewer, Tie1 antibody C-18 (Santa Cruz Biotechnology) is a polyclonal antibody raised in rabbits using the cytoplasmic peptide as an immunogen. The vendor's information states "Tie-1 (C-18) is recommended for detection of Tie-1 and, to lesser extent, Tie-2 of mouse, rat and human origin by Western blotting, immunoprecipitation, immunofluorescence and solid phase ELISA." Thus, the antibody cross-reacts with Tie2. We have included this information in the Materials and Methods section of the revised manuscript (page 11, line 315).

11. Based on the materials and methods, it is unclear in which media the cells were cultured for the Transwell assays and what is meant by starvation in this experiment.

We apologize for the insufficient description of the media used to culture the cells for the Transwell assay. The medium used for maintenance of LECs was Endothelial Cell Growth Medium (EGM)-MV2 (PromoCell) containing 5% FBS and the following growth factors: EGF (5 ng/ml), bFGF (10 ng/ml), IGF (20 ng/ml), and VEGF-A-165 (0.5 ng/ml). The LECs used in the Transwell assay were serum-starved overnight by replacing the EGM-MV2 containing FBS and growth factors with Endothelial Cell Basal Medium (EBM)-MV2 without FBS and growth factor supplementation. The above information has been included in the Materials and Methods section of the revised manuscript (page 14, lines 399–404).

12. It would be helpful to compare the treatments if molar concentrations are provided in Fig. 3A.

We have added the molar concentrations for Polydom, collagen, fibronectin, and laminin to the legend for **Fig. 3 A**, in accordance with the reviewer's suggestion.

Reviewer #3 (Comments to the Authors (Required)):

In this study the authors extend prior work on the role of the matrix protein Polydom during lymphatic vessel growth and development. Polydom is required for lymphatic vessel growth, maturation and valve formation, but the molecular mechanism by which it regulates lymphatic development remains unclear. Prior work by these authors identified Polydom as a high affinity ligand for integrin $\alpha 9\beta 1$, an adhesive receptor known to participate in lymphatic valve formation and vessel maturation. Subsequent studies in fish and mice established a conserved role for Polydom in lymphatic development. The authors also previously reported that Polydom binds Ang1 and 2, suggesting that its role in lymphatic growth may be related to Angiopoietin activity as well as $\alpha 9\beta 1$ adhesion. In the present study the authors identify binding of Polydom to Tie1 and use in vitro studies to demonstrate that a Polydom mutant unable to bind Tie1 is unable to stimulate LEC migration like wild-type Polydom. They conclude from these studies that Polydom is a Tie1 ligand and that its effects on LEC migration - and by extension lymphatic development? - are mediated by Tie1. Unfortunately, the data presented in the paper do not strongly support this conclusion and fail to explain (i) how Polydom-Tie1 binding controls migration and LEC function, and (ii) how Polydom binding to $\alpha 9\beta 1$ and Ang ligands contributes to its function if Tie1 binding is required. The study is very preliminary and requires more rigorous biochemical and functional studies to fully test the hypothesis that Polydom regulates lymphatic maturation and growth through Tie1 ligand activity.

Major points:

1. Does Polydom stimulate Tie1 signaling? The authors present binding data that support Polydom-Tie1 interaction and suggest that these findings identify Polydom as a novel Tie1 ligand that is required for lymphatic growth and maturation. Presumably such an effect would be mediated by Tie1 signal transduction, esp since they provide data excluding Tie2, but the authors fail to state this overtly and do not measure any signaling effects of Polydom. Studies testing Polydom stimulation of Tie1 phosphorylation and downstream signaling effectors such as ERK1/2 and PI3K/AKT/mTOR is essential to address their hypothesis.

We thank the reviewer for the supportive and constructive comments. We are fully aware of the importance of examining whether Polydom binding to Tie1 induces Tie1 phosphorylation and/or Tie2 cross-phosphorylation. Thus, we repeatedly performed Tie1/Tie2 phosphorylation assays by immunoprecipitation of Tie1/Tie2 from lysates of Polydom-treated LECs, followed by immunoblotting for tyrosine-phosphorylated Tie1/Tie2. However, we did not obtain any convincing evidence for Tie1 or Tie2 phosphorylation in Polydom-treated LECs (**new Supplementary Fig. S3 A**); we did observe a very small increase in phosphorylated Tie1 signals once or twice, but this small signal increase was not reproduced in more than 10 other independent experiments. We therefore concluded that Tie1 ligation by Polydom does not increase the phosphorylation of Tie1 or Tie2 in LECs. This conclusion was supported by another experiment in which 293-F cells were transfected with full-length Tie1 and/or Tie2, followed by Polydom treatment. Again, no significant Tie1 phosphorylation was induced by Polydom in the Tie1-transfected cells (**new Supplementary Fig. S3 B**). Notably, Tie1 phosphorylation was induced in cells co-transfected with Tie1 and Tie2 irrespective of the presence or absence of Polydom, consistent with previous observations that Tie1 was only phosphorylated in 293 cells when Tie1 and Tie2 were co-expressed (Saharinen et al., *J Cell Biol*, 2005;169:239–243; Yuan et al., *FASEB J*, 2007;21:3171–3183; Korhonen et al., *J Clin Invest*, 2016;126:3495–3510).

It is well known that the PI3K/Akt signaling pathway becomes activated downstream of the Ang-Tie axis. We therefore examined whether the PI3K/Akt pathway was activated in LECs following Tie1 ligation by Polydom. We found that Polydom induced Akt phosphorylation in LECs in a Tie1-

dependent manner (**new Fig. 4 B**), indicating involvement of the PI3K/Akt pathway in the Polydom-induced LEC migration. In support of this notion, Polydom-induced transmigration was inhibited by two PI3K inhibitors, LY294002 and Wortmannin, but not by an ERK inhibitor (**new Fig. 4 A**). In addition, we observed that nuclear exclusion of FOXO1, a signaling event elicited downstream of Akt activation, was induced in LECs after Polydom treatment (**new Fig. 4, C and D**). These new data have been added to the revised manuscript (page 7, lines 182–196).

2. Role of $\alpha 9\beta 1$ binding. The authors use the Polydom E2567A mutant as evidence that Polydom's effect on LEC migration is Tie1-mediated. Does the E2567A mutant impact integrin $\alpha 9\beta 1$ binding? If yes, their conclusion would appear over-simplified and not complete. If no, how do the authors reconcile their findings with prior work demonstrating strong $\alpha 9\beta 1$ binding and the fact that loss of both Polydom and $\alpha 9\beta 1$ result in lymphatic valve defects in vivo? The argument that $\alpha 9\beta 1$ is not the relevant receptor for Polydom because the phenotype of the Polydom mutant and the Itga9 mutant are not precisely identical is not a convincing one because there may be redundancy for integrin/adhesion receptor binding of Polydom.

We thank the reviewer for this comment. We examined the binding of integrin $\alpha 9\beta 1$ to the E2567A Polydom mutant and found that the E2567A mutation had no impact on Polydom binding to Tie1 (please see the data below).

In the original manuscript, we demonstrated that knockdown of Itga9 did not affect LEC migration in our Transwell assay (**Supplementary Fig. S2** in the original manuscript; renumbered as **Supplementary Fig. S5, A–C** in the revised manuscript), suggesting that integrin $\alpha 9\beta 1$ is not instrumental for lymphatic vessel remodeling. To further support this notion, we produced a Polydom mutant in which the integrin $\alpha 9\beta 1$ recognition sequence EDDMMEVPY was substituted with EDAMMAVPY. We found that the Polydom mutant was unable to bind to integrin $\alpha 9\beta 1$ but remained fully active in promoting LEC migration in our Transwell assay (**new Supplementary Fig. S5, D–F** and page 9, lines 255-258, in the revised manuscript). Taken together, these findings indicate that integrin $\alpha 9\beta 1$ is not instrumental for Polydom-induced LEC migration. Nevertheless, because of the defects in lymphatic valve formation in both Polydom-knockout and Itga9-knockout mice, we are aware of the importance of further investigating the mechanism for how Polydom and Itga9 cooperate in lymphatic valve formation and will continue to pursue this issue in a future study.

3. Role of Angiopoietin binding. The authors have also previously reported that Polydom binds Ang1 and Ang2, and argue in the discussion that this is not relevant to the present findings because Tie2 knockdown did not block Transwell migration of LECs stimulated by Polydom. This argument further suggests that they are proposing that Polydom stimulates Tie1 signaling, so direct signaling assays are essential. Also, is it possible that Tie1 binding is mediated by Polydom-bound Angiopoietin ligands rather than a direct Tie1-Polydom interaction? The fact that the authors have now identified 4 binding partners for Polydom makes this biology less rather than more transparent.

We thank the reviewer for raising this important question. Because Polydom binds to Ang2 (Figure 6 in Morooka et al., *Circ Res*, 2017;120:1276–1288) and Ang2 is secreted by LECs (Veikkola et al., *FASEB J*, 2003;17:2006–2023; Jang et al., *Arterioscler Thromb Vasc Biol*, 2009;29:401–407) as a homodimer and/or oligomer (Davis et al., *Nat Struct Biol*, 2003;10:38–44), it is tempting to speculate that Ang2 operates as a crosslinker of Polydom monomers and increases the avidity of the Polydom-Tie1 interaction, thereby enabling the signaling events elicited downstream of the interaction, including activation of the PI3K/Akt signaling pathway. We are attempting to narrow down the amino acid residues in Polydom to construct a Polydom mutant that is unable to bind to Ang2 but has no deteriorative impact on Tie1 binding. This type of Polydom mutant protein will provide further insights into the multimodal interactions of Polydom with Tie1, Ang2, and integrin $\alpha 9\beta 1$.

4. In vivo correlation. The authors provide no in vivo data to support the proposed Tie1 mechanism. The functional studies are limited to Transwell migration assays that may or may not be relevant to the major in vivo phenotypes observed with loss of Polydom in vivo. Supportive in vivo data would help clarify the mechanism.

We thank the reviewer for this comment. As detailed in our response to the reviewer's comment 1, Polydom activates the PI3K/Akt signaling pathway and induces nuclear exclusion of FOXO1. Therefore, we examined whether the activation/inactivation status of Foxo1 is altered in Polydom-deficient mice. We found that Foxo1 was exclusively localized within the nucleus in Polydom-deficient mice, while it was detected in both the nucleus and cytoplasm in wild-type mice, with significantly reduced signal intensity in the nucleus of the wild-type mice compared with the Polydom-deficient mice (**new Fig. 4, E and F**). These data provide in vivo evidence for the physiological importance of Polydom binding to Tie1 (page 7, lines 196–203).

Recently, Stefan Schulte-Merker and his group deposited a preprint in bioRxiv entitled "svpe1 and tie1 genetically interact and affect aspects of facial lymphatic development in a Vegf-independent manner" (Hußmann et al., 2022; <https://doi.org/10.1101/2022.09.28.509871>). This paper provides genetic evidence for an interaction of svpe1/polydom with tie1 based on similarities in the phenotypes of svpe1-knockout and tie1-knockout zebrafish, further supporting a physiological function for the Polydom-Tie1 interaction in lymphangiogenesis (page 10, line 288 through page 11, line 296).

Minor points:

Throughout the manuscript the authors fail to clearly state how they believe Polydom-Tie1 interaction regulates LEC migration at the molecular level. This needs to be stated in a clear and forthright manner, e.g. accompanied by a diagram that fully explains the authors' proposed molecular mechanism.

We thank the reviewer for this encouraging suggestion. Our findings clearly show that Polydom binds to Tie1 and activates the PI3K/Akt signaling pathway, followed by nuclear exclusion of Foxo1. However, it remains to be elucidated how Polydom activates the PI3K/Akt pathway without an apparent increase in Tie1/Tie2 phosphorylation and how Ang2 participates in these signaling events through binding to Polydom. Although knockdown of Tie2 does not compromise the Polydom-induced LEC migration, Tie2 is known to form a complex with Tie1 and to play central roles in blood vessel maturation and stabilization. Furthermore, integrins, particularly alpha5beta1 and alphavbeta3, may also participate in the endothelial cell responses to Ang1 and Ang2 (Cascone et al., *J Cell Biol*, 2005;170:993–1004; Hakanpaa et al., *Nat Commun*, 2015;6:5962, doi:10.1038/ncomms6962; Korhonen et al., *J Clin Invest*, 2016;126:3495–3510). We certainly need more data to draw a scheme for how Polydom activates the PI3K/Akt pathway downstream of the Ang-Tie signaling system and modulates lymphatic vessel remodeling and valve formation.

May 8, 2023

Re: JCB manuscript #202208047R-A

Prof. Kiyotoshi Sekiguchi
Osaka University
Institute for Protein Research
3-2 Yamadaoka
Suita, Osaka 565-0871
Japan

Dear Prof. Sekiguchi,

Thank you for submitting your manuscript entitled "Polydom/SVEP1 binds to Tie1 and promotes migration of lymphatic endothelial cells". The manuscript was assessed by expert reviewers, whose comments are appended to this letter. Although we still feel this work is suitable for JCB, some issues must be resolved at this stage.

You will see that reviewers were encouraged by the important improvements included in this revision. While they acknowledged that the mechanism of Tie1 activation remains unresolved, two reviewers were supportive of publication. Reviewer 3 requested confirmation that Polydom signaling through PI3K/Akt also requires Tie1 using Tie1 KO animals. We agree these data should be included, and in addition, while the in vivo data in Fig 4E and 4F is encouraging we feel that this observation should be supported with quantification. Please note that we will assess these additions without returning this manuscript to the reviewers.

GENERAL GUIDELINES:

Text limits: Character count for a Report is < 20,000, not including spaces. Count includes title page, abstract, introduction, the joint Results & Discussion, and acknowledgments. Count does not include materials and methods, figure legends, references, tables, or supplemental legends.

Figures: Reports may have up to 5 main text figures. To avoid delays in production, figures must be prepared according to the policies outlined in our Instructions to Authors, under Data Presentation, <https://jcb.rupress.org/site/misc/ifora.xhtml>. All figures in accepted manuscripts will be screened prior to publication.

Supplemental information: There are strict limits on the allowable amount of supplemental data. Reports may have up to 3 supplemental figures. Up to 10 supplemental videos or flash animations are allowed. A summary of all supplemental material should appear at the end of the Materials and methods section.

Please note that JCB now requires authors to submit Source Data used to generate figures containing gels and Western blots with all revised manuscripts. This Source Data consists of fully uncropped and unprocessed images for each gel/blot displayed in the main and supplemental figures. Since your paper includes cropped gel and/or blot images, please be sure to provide one Source Data file for each figure that contains gels and/or blots along with your revised manuscript files. File names for Source Data figures should be alphanumeric without any spaces or special characters (i.e., SourceDataF#, where F# refers to the associated main figure number or SourceDataFS# for those associated with Supplementary figures). The lanes of the gels/blots should be labeled as they are in the associated figure, the place where cropping was applied should be marked (with a box), and molecular weight/size standards should be labeled wherever possible. Source Data files will be made available to reviewers during evaluation of revised manuscripts and, if your paper is eventually published in JCB, the files will be directly linked to specific figures in the published article.

The typical timeframe for revisions is three to four months. While most universities and institutes have reopened labs and allowed researchers to begin working at nearly pre-pandemic levels, we at JCB realize that the lingering effects of the COVID-19 pandemic may still be impacting some aspects of your work, including the acquisition of equipment and reagents. Therefore,

if you anticipate any difficulties in meeting this aforementioned revision time limit, please contact us and we can work with you to find an appropriate time frame for resubmission. Please note that papers are generally considered through only one revision cycle, so any revised manuscript will likely be either accepted or rejected.

Thank you for this interesting contribution to Journal of Cell Biology. You can contact us at the journal office with any questions, cellbio@rockefeller.edu or call (212) 327-8588.

Sincerely,

Tatiana Petrova
Monitoring Editor
Journal of Cell Biology

Tim Fessenden
Scientific Editor
Journal of Cell Biology

Reviewer #1 (Comments to the Authors (Required)):

The authors have provided additional evidence to support their finding that the extracellular matrix protein Polydom functions as a ligand for the Tie1 receptor. In particular, their new data show that although Polydom does not induce phosphorylation of Tie1 in LECs, it induces Tie1-dependent activation of PI3K/Akt signaling in cultured LECs. They also observed increased nuclear localization of FOXO1 in lymphatic vessels of Polydom deficient mice, consistent with a decrease in PI3K signaling. Although the physiological function for the Polydom-Tie1 interaction in lymphangiogenesis in vivo has not been addressed in the current study, the study by Hußmann et al, recently published in eLife (PMID: 37097004), provides compelling in vivo evidence (but also biochemical evidence) for Svep1/ Polydom-Tie1 interaction and its role in regulating lymphangiogenesis in zebrafish. As such, the current study is a valuable and timely independent contribution to the significant discovery of Polydom as a Tie1 ligand. Yet, it would have been of interest to characterize the suggested Polydom-Ang2-Tie-integrin signaling mechanism in the current study.

Specific comments:

1. The authors should avoid statements that are not based on data presented in the manuscript. For example, in the abstract, I suggest revising the following sentence, and focusing on current findings: "Given the similarities between the phenotypes of Polydom null and Tie1-null mice, these findings indicate that Polydom is a physiological ligand for Tie1..."
2. It would be appropriate to discuss the study by Hußmann et al (PMID: 37097004) already in the introduction.

Reviewer #2 (Comments to the Authors (Required)):

The authors have responded to the comments and included new data.

The revised manuscript shows that Polydom binds to Tie1 but does not induce its phosphorylation (new Fig S3). Polydom induced the PI3K/Akt pathway (new Fig. 4A-B), and LEC migration in a Tie1 and PI3K-dependent, but alpha9-integrin (new Fig S5D-F) and Tie1/Tie2 phosphorylation-independent manner (new Fig S3). Nuclear FOXO1 was decreased in Polydom-treated cultured LECs (new Fig. 4C-D) but increased in the lymphatic vessels of Polydom deficient mice (new Fig 4E, Fig S4).

Main comments:

1. The new data links Polydom signaling to FOXO1 both in vitro and in vivo, but the mechanism how Polydom induces downstream signaling remains unclear.
2. As Polydom did not induce Tie1/Tie2 phosphorylation, the mechanism by which Tie1 mediates Polydom signaling remains

unresolved.

3. The authors provide evidence supporting correct folding of the E2567A mutant, but do not e.g. directly show its functionality using other assays than Tie1 binding.

4. New analysis of LEC adhesion and migration provides somewhat controversial results: polydom inhibited LEC adhesion (new Fig S2A) as well as LEC migration, when Polydom was attached to the bottom of the Transwell filter, whereas the addition of Polydom to the lower chamber under these conditions had no effect (new Fig. S2B-C). Moreover, VEGF-C and Polydom failed to stimulate LEC migration in the Transwell and scratch-wound assays, respectively (comment to reviewer 1).

5. An RGD containing peptide inhibited Polydom-mediated LEC migration (new Fig S2D). However, the role of RGD binding integrins in polydom signaling remains to be shown.

6. The authors should include details of statistical analysis in the manuscript and check if pairwise comparison test is suitable, e.g. in new Fig 4D, containing multiple groups.

Reviewer #3 (Comments to the Authors (Required)):

The main weakness in the first submission of this manuscript was the lack of data demonstrating a clear role for Polydom as a signal-transducing ligand for Tie1. The authors have added additional signaling data to test their contention that Polydom is a Tie1 ligand that drives endothelial responses through Tie1 mediated signals. Unfortunately, these studies fail to make a convincing case for this conclusion. On the positive side, they now report PI3K signaling in response to Polydom in cultured ECs. Although this is also observed with use of a Polydom mutant unable to bind Tie1, they fail to show that it is either Tie1-dependent, e.g. by knocking down Tie1 in ECs, or mediated by Tie1 signaling. Thus there remains no direct connection between the induction of PI3K signaling and the observed biochemical association with Tie1 in the revised studies.

The authors also fail to demonstrate any in vivo data that support their conclusions. Instead, they refer to a Schulte-Merker lab publication on BioRxiv that identifies a genetic interaction in zebrafish, an association that also fails to directly support their contention Polydom as a Tie1 ligand that induces relevant signal transduction.

Overall my impression is that the conclusions in this manuscript mostly rely upon biochemical associations that are not adequately supported either by in vitro studies of endothelial cell signaling or by in vivo studies. Given the relatively large number of putative molecular mechanisms put forth by the authors to explain Polydom function based on similar biochemical approaches (i.e. as a binding partner for integrin $\alpha 9 \beta 1$, Ang1, Ang2 and now Tie1), these studies fail to clearly advance the molecular and cellular understanding of Polydom function.

Reviewer #1 (Comments to the Authors (Required)):

Specific comments:

1. *The authors should avoid statements that are not based on data presented in the manuscript. For example, in the abstract, I suggest revising the following sentence, and focusing on current findings: "Given the similarities between the phenotypes of Polydom null and Tie1-null mice, these findings indicate that Polydom is a physiological ligand for Tie1..."*

According to the reviewer's suggestion, we rephrased the sentence in the abstract as follows:

~~Given the similarities between the phenotypes of Polydom null and Tie1 null mice,~~ These findings indicate that Polydom is a physiological ligand for Tie1 and participates in lymphatic vessel development through activation of the PI3K/Akt pathway.

2. *It would be appropriate to discuss the study by Hußmann et al (PMID: 37097004) already in the introduction.*

At the time of the submission of our first manuscript, Hußmann et al (PMID: 37097004) was not submitted yet. Our study was completed without knowing the work by Hußmann et al and therefore fully independent of their study on zebrafish. We are happy to discuss their work in Discussion, but we believe that we do not need to refer to their work as one of the background information.

Reviewer #2 (Comments to the Authors (Required)):

Main comments:

1. *The new data links Polydom signaling to FOXO1 both in vitro and in vivo, but the mechanism how Polydom induces downstream signaling remains unclear.*
2. *As Polydom did not induce Tie1/Tie2 phosphorylation, the mechanism by which Tie1 mediates Polydom signaling remains unresolved.*

As pointed out by the reviewer, the mechanism of how Polydom induces downstream signaling remains unresolved. We will continue to address the mechanism by focusing on the interplay between the Ang/Tie system and integrins.

3. The authors provide evidence supporting correct folding of the E2567A mutant, but do not e.g. directly show its functionality using other assays than Tie1 binding.

In addition to the Tie1 binding assays, we examined the binding of the E2567A mutant to integrin alpha9beta1. As shown below, we confirmed that the E2567A mutation did not compromise the Polydom binding to integrin alpha9beta1. AMMA stands for the polydom mutant in which the integrin alpha9beta1 binding motif EDDMMVEVPY was substituted with EDAMMAVPY to nullify the integrin binding activity.

4. New analysis of LEC adhesion and migration provides somewhat controversial results: polydom inhibited LEC adhesion (new Fig S2A) as well as LEC migration, when Polydom was attached to the bottom of the Transwell filter, whereas the addition of Polydom to the lower chamber under these conditions had no effect (new Fig. S2B-C). Moreover, VEGF-C and Polydom failed to stimulate LEC migration in the Transwell and scratch-wound assays, respectively (comment to reviewer 1).

The question asked in the experiments shown in **Fig. S2** is the mechanism by which Polydom promotes LEC migration, i.e., whether it is induced in a chemotactic mechanism or in a haptotactic mechanism. The results shown in **Fig. S2A** does not show that Polydom inhibits LEC adhesion. It shows that Polydom is much less potent than collagen, fibronectin, and laminin in promoting cell-substrate adhesion, as evidenced by the reduced number of attached cells and less prominent

spreading of the attached cells. Notably, collagen, fibronectin, and laminin, all of which possess more potent cell-adhesive activity than Polydom, do not induce LEC migration when added to the lower chamber medium in Transwell migration assays (**Fig. 3A**), implying that Polydom-induced LEC migration is not driven by a haptotactic mechanism but driven by a chemotactic mechanism. In support of this view, LEC transmigration was not induced when the lower surface of the Transwell membrane was precoated with Polydom and then blocked with BSA, even though Polydom was added to the lower chamber medium as an attractant (**Fig. S2B-C**). Thus, Polydom does not seem to function as a cell-adhesive substrate. Because the lower chamber medium contains 0.5% fetal bovine serum, which has been known to contain fibronectin and vitronectin, the well characterized RGD-containing cell-adhesive proteins. It is therefore conceivable that such serum proteins are adsorbed onto the lower surface of the Transwell membrane in our Transwell assays and provide adhesive substrates for LECs that are attracted by Polydom added to the lower chamber medium. This scenario has been supported by the inhibition of LEC transmigration by the RGD peptide added to the lower chamber medium (**Fig. S2D**).

5. An RGD containing peptide inhibited Polydom-mediated LEC migration (new Fig S2D). However, the role of RGD binding integrins in polydom signaling remains to be shown.

As pointed out by the reviewer, the role of RGD binding integrins in Polydom-induced signaling remains to be elucidated. We will pursue this issue in our future study.

6. The authors should include details of statistical analysis in the manuscript and check if pairwise comparison test is suitable, e.g. in new Fig 4D, containing multiple groups.

We revised **Fig. 4D** to facilitate pairwise comparison among different groups of cells showing distinct FOXO1 localization (new **Fig. 4D**). The details of statistical analysis have been described in **Materials and Methods** (page 17, lines 481-484). We also quantified the results shown in **Fig. 4E** and **4F**, which show the nuclear localization of Foxo1 in wild-type and Polydom-deficient mice, by counting the number of cells showing distinct Foxo1 localization. The results have now been included as the new **Fig. 4F**.

Reviewer #3 (Comments to the Authors (Required)):

The main weakness in the first submission of this manuscript was the lack of data demonstrating a clear role for Polydom as a signal-transducing ligand for Tie1. The authors have added additional signaling data to test their contention that Polydom is a Tie1 ligand that drives endothelial responses through Tie1 mediated signals. Unfortunately, these studies fail to make a convincing case for this conclusion. On the positive side, they now report PI3K signaling in response to Polydom in cultured ECs. Although this is also observed with use of a Polydom mutant unable to bind Tie1, they fail to show that it is either Tie1-dependent, e.g. by knocking down Tie1 in ECs, or mediated by Tie1 signaling. Thus there remains no direct connection between the induction of PI3K signaling and the observed biochemical association with Tie1 in the revised studies.

Our conclusion that Polydom-induced PI3K signaling and subsequent LEC migration is dependent on Polydom binding to Tie1 is based on our following findings. First, the E2567A Polydom mutant that is unable to bind Tie1 does not activate Akt phosphorylation as shown in **Fig. 4B**. Second, the E2567A mutation compromises the Polydom-induced LEC migration in our Transwell assays as shown in **Fig. 3G** and **3H**. We believe that these results support our conclusion that Polydom-induced PI3K signaling as well as LEC migration is Tie1-dependent.

We noticed that we made an error in our response to the second comment of Reviewer #3, which might have confused Reviewer #3 in reviewing our revised manuscript. In the comment, Reviewer #3 asked whether the E2567A mutation may have an impact on Polydom binding to integrin alpha9beta1. We therefore performed integrin binding assays with the E2567A mutant and confirmed that the E2567A mutation did not compromise the integrin alpha9beta1 binding activity of Polydom. However, in the response to this comment, we wrote "the E2567A mutation had no impact on Polydom binding to **Tie1**", instead of "the E2567A mutation had no impact on Polydom binding to **integrin alpha9beta1**". Reviewer #3 might have understood that "the E2567A mutation had no impact on Polydom binding to **Tie1**" as written in our response. We deeply apologize for the erroneous description in our response to the reviewer's comment.

The authors also fail to demonstrate any in vivo data that support their conclusions. Instead, they refer to a Schulte-Merker lab publication on BioRxiv that identifies a genetic interaction in zebrafish, an association that also fails to directly support their contention Polydom as a Tie1 ligand that induces relevant signal transduction.

A revised version of the Schulte-Merker's paper has now been published in eLife (<https://elifesciences.org/articles/82969>). In the revised paper, they provide not only genetic evidence for the Polydom-Tie1 association but also biochemical evidence by performing coimmunoprecipitation experiments, further supporting our conclusion that Polydom is a physiological ligand for Tie1.

Overall my impression is that the conclusions in this manuscript mostly rely upon biochemical associations that are not adequately supported either by in vitro studies of endothelial cell signaling or by in vivo studies. Given the relatively large number of putative molecular mechanisms put forth by the authors to explain Polydom function based on similar biochemical approaches (i.e. as a binding partner for integrin $\alpha 9\beta 1$, Ang1, Ang2 and now Tie1), these studies fail to clearly advance the molecular and cellular understanding of Polydom function.

As pointed out by the reviewer, Polydom interacts with multiple players involved in blood and lymphatic vessel development, i.e., Tie1, integrin $\alpha 9\beta 1$, Ang1, and Ang2, raising the possibility that Polydom functions as a platform that orchestrates the functions of these angiogenic/lymphangiogenic factors. Although further investigation is needed to elucidate the function of Polydom in lymphangiogenesis, we believe that our findings that Polydom binds to Tie1, a long-standing orphan receptor in the Ang-Tie system, and activates PI3K/Akt signaling pathway is an important advance in understanding how blood and lymphatic vessel development is regulated by the Ang-Tie system.

May 22, 2023

RE: JCB Manuscript #202208047RR

Prof. Kiyotoshi Sekiguchi
Osaka University
Institute for Protein Research
3-2 Yamadaoka
Suita, Osaka 565-0871
Japan

Dear Prof. Sekiguchi:

Thank you for submitting your revised manuscript entitled "Polydom/SVEP1 binds to Tie1 and promotes migration of lymphatic endothelial cells". We would be happy to publish your paper in JCB pending final revisions necessary to meet our formatting guidelines (see details below).

A. MANUSCRIPT ORGANIZATION AND FORMATTING:

Full guidelines are available on our Instructions for Authors page, <http://jcb.rupress.org/submission-guidelines#revised>. Submission of a paper that does not conform to JCB guidelines will delay the acceptance of your manuscript.

1) Text limits: Character count for Articles is < 40,000, not including spaces. Count includes abstract, introduction, results, discussion, and acknowledgments. Count does not include title page, figure legends, materials and methods, references, tables, or supplemental legends.

2) Figures limits: Articles may have up to 10 main figures and 5 supplemental figures/tables.

3) Figure formatting: Scale bars must be present on all microscopy images, including inset magnifications. Molecular weight or nucleic acid size markers must be included on all gel electrophoresis.

4) Statistical analysis: Error bars on graphic representations of numerical data must be clearly described in the figure legend. The number of independent data points (n) represented in a graph must be indicated in the legend. Statistical methods should be explained in full in the materials and methods. For figures presenting pooled data the statistical measure should be defined in the figure legends. Please also be sure to indicate the statistical tests used in each of your experiments (either in the figure legend itself or in a separate methods section) as well as the parameters of the test (for example, if you ran a t-test, please indicate if it was one- or two-sided, etc.). Also, if you used parametric tests, please indicate if the data distribution was tested for normality (and if so, how). If not, you must state something to the effect that "Data distribution was assumed to be normal but this was not formally tested."

** Please indicate n for all assays in all figure legends.

** Please indicate whether t-tests were one- or two-sided. This may be simpler to include in figure legends rather than in the methods section.

5) Abstract and title: The abstract should be no longer than 160 words and should communicate the significance of the paper for a general audience. The title should be less than 100 characters including spaces. Make the title concise but accessible to a general readership.

** Please consider including the observations in Figure S3 on Tie1 phosphorylation in the abstract. Although this remains an unresolved point, this finding would be important to convey to interested readers and merits inclusion in the abstract.

6) Materials and methods: Should be comprehensive and not simply reference a previous publication for details on how an experiment was performed. Please provide full descriptions in the text for readers who may not have access to referenced manuscripts. We also provide a report from SciScore and an associate score, which we encourage you to use as a means of evaluating and improving the methods section.

** Please indicate the methods used for mouse studies included in the revision.

7) Please be sure to provide the sequences for all of your primers/oligos and RNAi constructs in the materials and methods. You must also indicate in the methods the source, species, and catalog numbers (where appropriate) for all of your antibodies. Please also indicate the acquisition and quantification methods for immunoblotting/western blots.

8) Microscope image acquisition: The following information must be provided about the acquisition and processing of images:

- Make and model of microscope
- Type, magnification, and numerical aperture of the objective lenses
- Temperature
- Imaging medium
- Fluorochromes
- Camera make and model
- Acquisition software
- Any software used for image processing subsequent to data acquisition. Please include details and types of operations involved (e.g., type of deconvolution, 3D reconstitutions, surface or volume rendering, gamma adjustments, etc.).

10) Supplemental materials: There are strict limits on the allowable amount of supplemental data. Articles may have up to 5 supplemental figures. Please also note that tables, like figures, should be provided as individual, editable files. A summary of all supplemental material should appear at the end of the Materials and methods section.

13) ORCID IDs: ORCID IDs are unique identifiers allowing researchers to create a record of their various scholarly contributions in a single place. At resubmission of your final files, please consider providing an ORCID ID for as many contributing authors as possible.

Please note that JCB now requires authors to submit Source Data used to generate figures containing gels and Western blots with all revised manuscripts. This Source Data consists of fully uncropped and unprocessed images for each gel/blot displayed in the main and supplemental figures. Since your paper includes cropped gel and/or blot images, please be sure to provide one Source Data file for each figure that contains gels and/or blots along with your revised manuscript files. File names for Source Data figures should be alphanumeric without any spaces or special characters (i.e., SourceDataF#, where F# refers to the associated main figure number or SourceDataFS# for those associated with Supplementary figures). The lanes of the gels/blots should be labeled as they are in the associated figure, the place where cropping was applied should be marked (with a box), and molecular weight/size standards should be labeled wherever possible.

Journal of Cell Biology now requires a data availability statement for all research article submissions. These statements will be published in the article directly above the Acknowledgments. The statement should address all data underlying the research presented in the manuscript. Please visit the JCB instructions for authors for guidelines and examples of statements at (<https://rupress.org/jcb/pages/editorial-policies#data-availability-statement>).

B. FINAL FILES:

Thank you for this interesting contribution, we look forward to publishing your paper in Journal of Cell Biology.

Sincerely,

Tatiana Petrova
Monitoring Editor
Journal of Cell Biology

Tim Fessenden
Scientific Editor
Journal of Cell Biology